# A *PRDX1* mutant allele causes a *MMACHC* secondary epimutation in *cblC* patients

Jean-Louis Guéant[1], Céline Chéry[1], Abderrahim Oussalah [1], Javad Nadaf[2], David Coelho [1], Thomas Josse[1], Justine Flayac[1], Aurélie Robert[1], Isabelle Koscinski[1], Isabelle Gastin[1], Pierre Filhine-Tresarrieu[1], Mihaela Pupavac[2], Alison Brebner[2], David Watkins[2], Tomi Pastinen[2], Alexandre Montpetit[2], Fadi Hariri[2], David Tregouët[3], Benjamin A Raby[4], Wendy K. Chung[5], Pierre-Emmanuel Morange[6], D.Sean Froese[7], Matthias R. Baumgartner[7], Jean-François Benoist[8], Can Ficicioglu[9], Virginie Marchand[10], Yuri Motorin[10], Chrystèle Bonnemains[1], François Feillet[1], Jacek Majewski [2] & David S. Rosenblatt[2]

To date, epimutations reported in man have been somatic and erased in germlines. Here, we identify a cause of the autosomal recessive *cblC* class of inborn errors of vitamin $B_{12}$ metabolism that we name "epi-*cblC*". The subjects are compound heterozygotes for a genetic mutation and for a promoter epimutation, detected in blood, fibroblasts, and sperm, at the *MMACHC* locus; 5-azacytidine restores the expression of *MMACHC* in fibroblasts. *MMACHC* is flanked by *CCDC163P* and *PRDX1*, which are in the opposite orientation. The epimutation is present in three generations and results from *PRDX1* mutations that force antisense transcription of *MMACHC* thereby possibly generating a H3K36me3 mark. The silencing of *PRDX1* transcription leads to partial hypomethylation of the epiallele and restores the expression of *MMACHC*. This example of epi-*cblC* demonstrates the need to search for compound epigenetic-genetic heterozygosity in patients with typical disease manifestation and genetic heterozygosity in disease-causing genes located in other gene trios.

[1] INSERM, UMR_S954 Nutrition-Genetics-Environmental Risk Exposure and Reference Centre of Inborn Metabolism Diseases, University of Lorraine and University Hospital Centre of Nancy (CHRU Nancy), 54505 Nancy, France. [2] Department of Human Genetics, McGill University and Research Institute McGill University Health Centre, Montreal H4A 3J1 Quebec, Canada. [3] Sorbonne Universités, UPMC University Paris 06, Institut National pour la Santé et la Recherche Médicale (INSERM), ICAN Institute for Cardiometabolism and Nutrition, Unité Mixte de Recherche en Santé (UMR_S) 1166, Team Genomics & Pathophysiology of Cardiovascular Diseases, 75013 Paris, France. [4] Channing Division of Network Medicine, Department of Medicine, Brigham and Women's Hospital, Harvard Medical School, Boston, MA 02115, United States of America. [5] Departments of Pediatrics and Medicine, Columbia University New York, NY 10032, United States of America. [6] INSERM, UMR_S1062, Nutrition Obesity and Risk of Thrombosis, Aix-Marseille University, 13005 Marseille, France. [7] Division of Metabolism and Children's Research Centre (CRC), University Children's Hospital, CH-8032 Zürich, Switzerland. [8] Service de Biochimie Hormonologie, Hôpital Robert Debré, 75019 Paris, France. [9] Children's Hospital of Philadelphia, Perelman School of Medicine at the University of Pennsylvania, Philadelphia, PA 19104, United States of America. [10] Laboratoire Ingénierie Moléculaire et Physiopathologie Articulaire (IMoPA), UMR7365 CNRS - Université de Lorraine and FR3209 CNRS- Université de Lorraine, 54505 Nancy, France. Correspondence and requests for materials should be addressed to J.-L.Géa. (email: jean-louis.gueant@univ-lorraine.fr)

Epigenetic diseases are caused by stable alterations of DNA methylation, posttranslational histone modification, and/or production of non-coding RNA. These phenotypic traits may be transmitted from parents to offspring. They are termed epimutations when they are directly involved in the underlying molecular mechanisms of the disease. Epimutations can be separated into two types, primary and secondary, the latter occurring secondary to a DNA mutation in a *cis*- or *trans*-acting factor[1, 2]. In most cases, epimutations are somatic and are likely to exist as mosaics with tissue-specific effects[2–4]. To our knowledge, to date there is no evidence of epimutations maintained in germlines in any of these cases[2]. For example, an epimutation reported in three generations with a familial cancer syndrome caused by epigenetic silencing of the *MLH1* gene is erased in spermatozoa, but reinstated in the somatic cells of the next generation[3, 5].

Here, we report cases with a rare inborn error of metabolism produced by the inheritance of a gene mutation that disrupts the gene function of a flanking gene through epigenetic mechanisms in somatic and germ line cells. The cases were classified as belonging to the autosomal recessive *cblC* (cobalamin, Cbl) class of inborn errors of vitamin $B_{12}$ metabolism, usually caused by homozygous or compound heterozygous mutations in the *MMACHC* gene[6, 7]. *MMACHC* encodes a protein with both chaperone and enzyme functions, and its inactivation disrupts the synthesis of two Cbl derivatives, methylcobalamin (MeCbl) and adenosylcobalamin (AdoCbl), which serve as cofactors for methionine synthase and methylmalonyl-CoA mutase, respectively[6, 7]. The *cblC* disorder presents with both severe neurologic and systemic metabolic abnormalities[6]. Here, we report a new cause of the *cblC* class of inborn errors of vitamin $B_{12}$ metabolism that we name "epi-*cblC*". The epi-*cblC* cases are compound heterozygous for a genetic mutation and a secondary epimutation at the *MMACHC* locus. The secondary epimutation was triggered by variants in the neighboring 3′ antisense gene *PRDX1* that produces an aberrant antisense transcript by skipping the polyadenylation site.

## Results

**Identification of a new cause of *cblC* named epi-*cblC*.** We initially identified the *MMACHC* secondary epimutation in two infant females with paternal transmission, case CHU-12122 (died at 1 month of age, Caucasian ancestry) and case WG-3838 (died at 2 months of age, Japanese and Korean ancestry) and in a 59-year-old male (case WG-4152, Caucasian ancestry). The *cblC* diagnosis was based on metabolic and fibroblast complementation analyses. Patient clinical and metabolic data are detailed in the Methods section.

Sanger sequencing of *MMACHC* in DNA of CHU-12122 revealed a single heterozygous *c.270_271insA*, p.Arg91LysfsX14 mutation, which was also identified in her unaffected mother and maternal grandfather (Fig. 1a). The maternal grandfather and the father had serum concentrations of homocysteine of 33.6 and 17.0 μmol/L, respectively (reference levels <15 μmol/L). No sequence variant in *MMACHC* was identified in the father and no additional gene mutations were identified in the patient and her relatives using the TruSight NGS Illumina panel of the genes involved in metabolic diseases. We performed methylation analysis of the *MMACHC* gene by PCR amplification and cloning, followed by Sanger sequencing of bisulfite-treated DNA. This revealed a heterozygous epimutation consisting of 32 hypermethylated CpG sites in a CpG island encompassing the promoter and first exon of *MMACHC* (Chr1:45,965,587–45,966,049) (Fig. 1b, c). The epimutation was observed in fibroblast DNA from the patient, blood and sperm

DNA from her father, and blood DNA from her paternal grandfather. We detected a *c.-302G* allele of the rs3748643 bearing the epimutation and a *c.-302T* allele bearing the non-methylated allele. The epimutation and *c.-302G* allele of rs3748643 were absent in the DNA from the mother and maternal grandmother (Fig. 1a, c). The *MMACHC* allele bearing the *c.270_271insA* mutation was the only one detected by RT-PCR analysis of patient fibroblast transcripts. Treating the fibroblasts with 5-azacytidine (5-AZA), an inhibitor of DNA methyltransferases (DNMTs), produced bi-allelic *MMACHC* expression (Fig. 1d). Taken together, these results showed an epimutation causing promoter hypermethylation and *MMACHC* silencing in the paternally transmitted allele and a *c.270_271insA* mutation in the allele inherited from the mother. The epimutation was present in three generations; it was transmitted to CHU-12122 by her father, and to her father by her grandfather and was present in the sperm of CHU-12122's father.

Sanger sequencing of *MMACHC* in blood DNA of WG-3838 revealed the presence of a previously reported heterozygous *c.81G>A* splice variant in the patient, her mother, and a deceased brother (Fig. 2a)[8]. RT-PCR of fibroblast transcripts found no detectable *MMACHC* cDNA. No paternal *MMACHC* mutation was identified. Sequencing of bisulfite-converted DNA identified the same mono-allelic epimutation as that identified in CHU-12122 in blood DNA samples from her and her father and in sperm from her father; the patient was compound heterozygous for the *c.81G>A* mutation and the *MMACHC* epimutation (Fig. 2b). In contrast to CHU-12122, the *c.-302T* allele of the rs3748643 *c.-302 G>T* polymorphism was detected with the epimutation and the *c.-302G* allele tagged the non-methylated allele. Taken together, these data showed a presentation similar to case CHU-12122. The epimutation silenced *MMACHC* gene expression in the fibroblasts.

Sequencing of the *MMACHC* gene in blood DNA of WG-4152 identified a previously described heterozygous *c.158T>C* (p. Leu53Pro) mutation[9]. Sequencing of bisulfite-converted DNA identified the same mono-allelic epimutation found in CHU-12122 and WG-3838. The patient was compound heterozygous for the *c.158T>C* mutation and the *MMACHC* epimutation. The *c.-302G* allele of rs3748643 was detected in the allele bearing the epimutation and the *c.-302T* allele in the non-methylated allele.

**Confirmation of the epimutation by epigenome-wide analysis.** We performed an epigenome-wide study of DNA from the 3 epi-*cblC* cases and their relatives using the Infinium HumanMethylation450 BeadChip array (HM450K)[10]. The methylation variation positions (MVPs) of the three cases were compared to those observed in 690 DNA samples from a Caucasian population from the Asthma BRIDGE consortium (USA)[11], 350 DNA samples from a Caucasian population recruited in the South of France (the MARTHA cohort)[12], and one typical *cblC* (CHU-09011) sample with compound heterozygosity for the same mutation as CHU-12122 in exon 2 (*c.270_271insA*, p.Arg91 > LysfsX14) and an exon 4 mutation (*c.616C>T*, p.Arg206Trp). The HM450K array included eight probes covering the CpG sites of the epimutation (Fig. 1b). HM450K methylome profiling in the three epi-*cblC* cases and their relatives confirmed the results obtained by Sanger sequencing of bisulfite-treated DNA (Fig. 2c, Fig. 2d). The epimutation was also confirmed in sperm from the fathers of the CHU-12122 and WG-3838 cases. The CpG hypomethylation in sperm DNA from a control population from Utah[13] was similar to that observed in the genomic DNA of the control subjects (Fig. 2e). We observed only one subject in the control population with the same hypermethylation of the *MMACHC* sites as was reported in the epi-*cblC* cases and their relatives. The allele

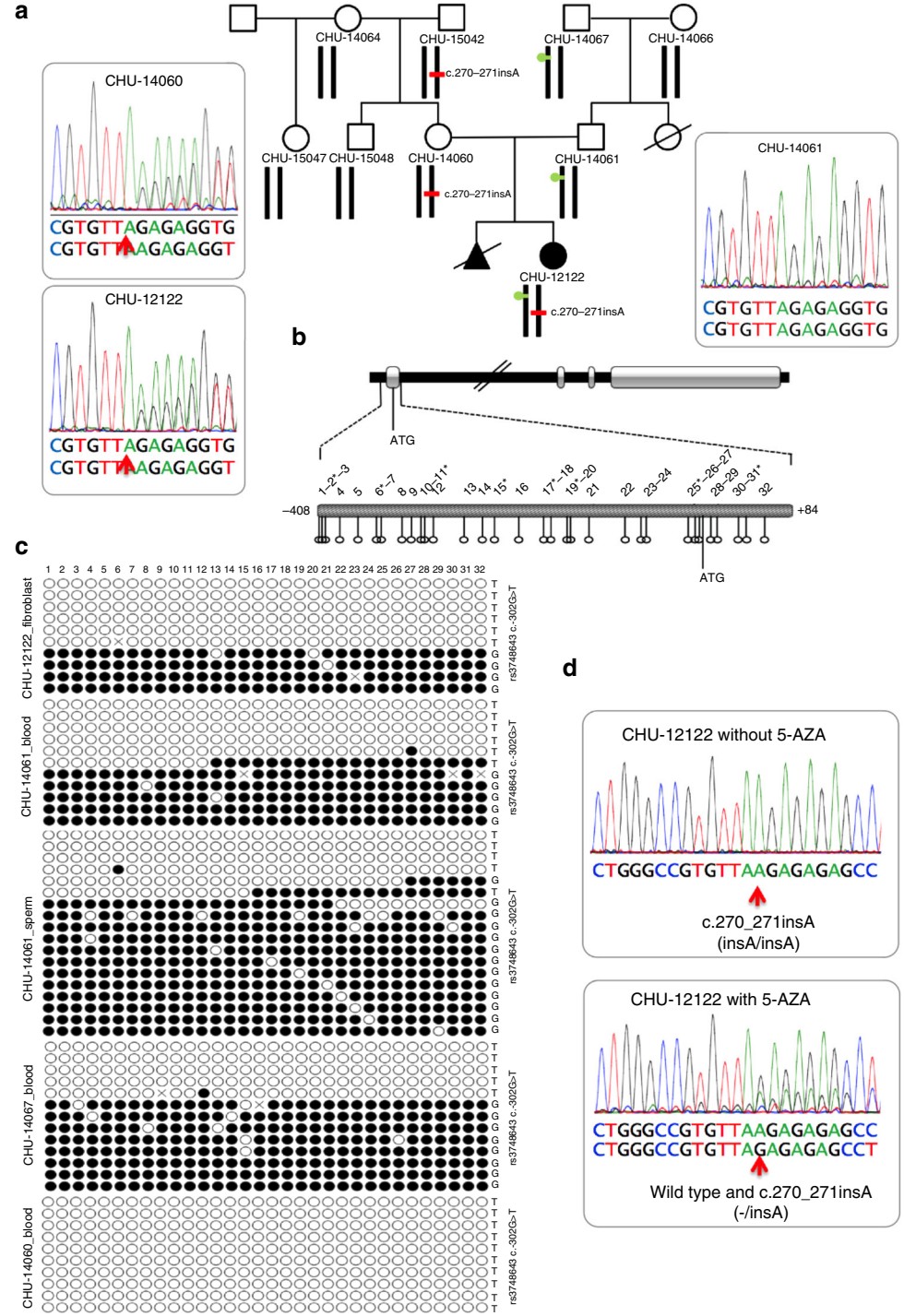

**Fig. 1** The CHU-12122 epi-*cblC* case is compound heterozygous for a coding mutation and an epimutation of *MMACHC* detected in three generations. **a** Pedigree of the family of case CHU-12122; red bar, mutation (heterozygous c.270_271insA, p.Arg91LysfsTer14) found in the proband, her mother, and her maternal grandfather; green circle, epimutation encompassing the *MMACHC* promotor/exon 1 found in the proband. **b** Map of the *MMACHC* gene and expanded view of the *MMACHC* CpG island. CpG sites are numbered according to their position upstream and downstream of ATG. Asterisks indicate the CpG sites of the CpG island which were probed in the HM450K array. **c** Epigrams of *MMACHC* methylation analyzed by PCR amplification/cloning/Sanger sequencing of bisulfite-treated DNA. Epimutations were detected in the proband, her father, and her paternal grandfather. **d** RT-PCR of fibroblasts from case CHU-12122 before and after the 3-day treatment with 10 μM 5-azacytidine (5-AZA). Silencing of the wild-type allele (mono-allelic expression in upper panel, red arrow) was reversed after treatment (bi-allelic expression after treatment, lower panel, red arrow)

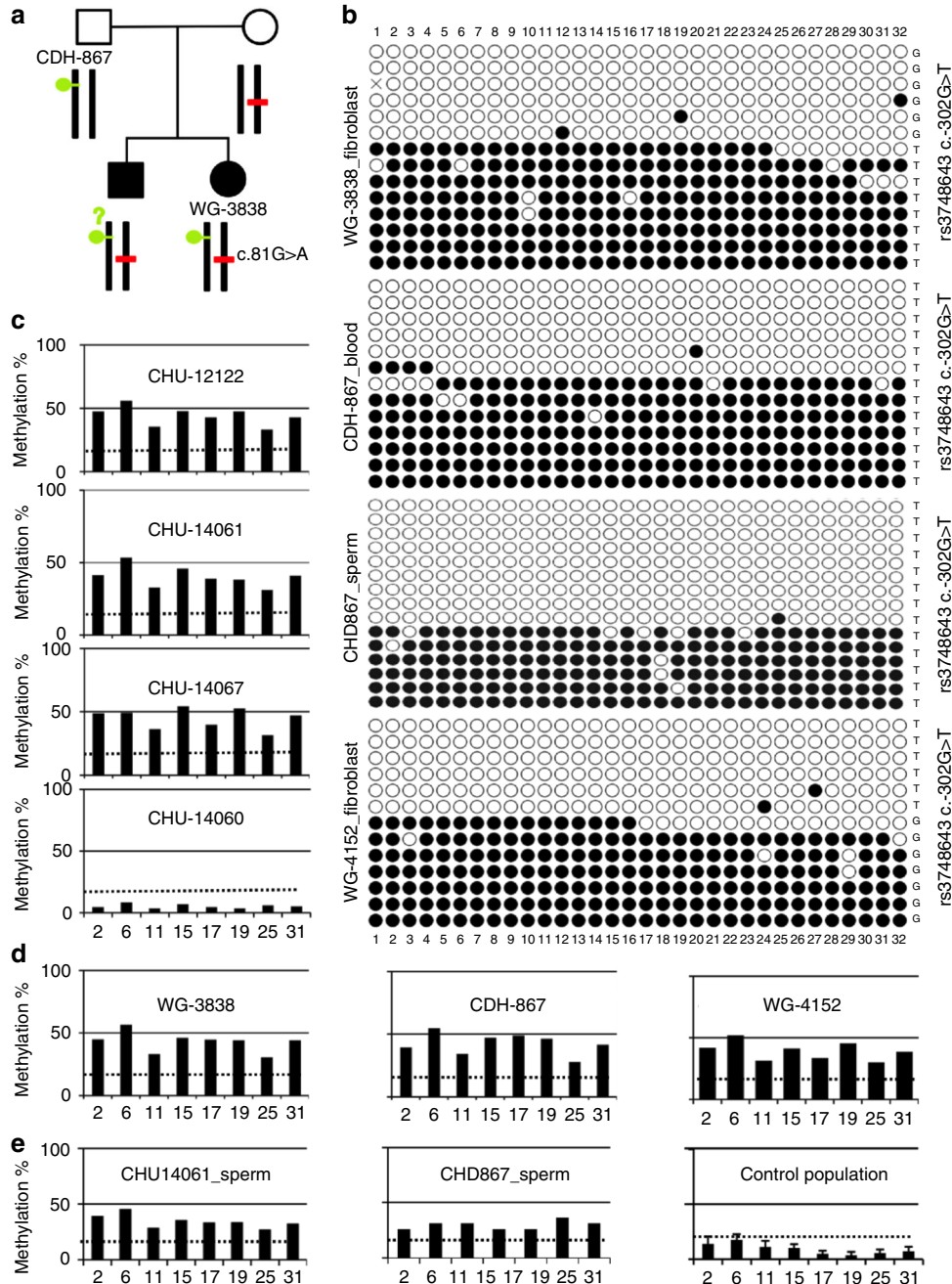

**Fig. 2** The *MMACHC* epimutation is found in a second case and is present in sperm of both cases. **a** Pedigree of the family of case WG3838; red bar, mutation (heterozygous c.81G>A splice variant) found in the proband, her mother, and a deceased brother; green circle, epimutation encompassing the *MMACHC* promotor/exon 1 found in the proband and her mother. No material from the deceased brother was available to study the presence of the epimutation. **b** *MMACHC* methylation epigrams after PCR amplification/cloning/Sanger sequencing of the bisulphite-treated DNA from blood obtained from WG-3838, her father CHD867, WG-4152, and from sperm obtained from CHD867. The epimutation was identified in all samples. **c** HM450K array methylome profiling of the epimutation in the CHU-12122 case and her relatives. The data confirmed the results obtained in Fig. 1c. **d** Methylome profiling confirms the epimutation in blood DNA from case WG-3838, her father CDH-867, and case WG-4152; the dotted line corresponds to a β value threshold of 0.2, below which the CpG probe was considered fully unmethylated. **e** Methylome profiling of sperm DNA from the father of case CHU-12122, the father of case WG-3838, and a control population of Utah. The data confirm the epimutation's presence in the sperm; the dotted line corresponds to a β value threshold of 0.2, below which the CpG probe was considered fully unmethylated. The absence of sperm DNA contamination by DNA from somatic cells was proven by methylation analysis of SNRPN imprinted gene (see Supplementary Fig. 1)

frequency of the epimutation was therefore not higher than $5 \times 10^{-4}$ in the control population. We also confirmed the absence of the *MMACHC* epimutation in the CHU-09011 *cblC* case.

To compare the epigenome-wide changes between the three cases and their relatives with those from the control population, we plotted the difference between the beta methylation coefficients of the MVPs in cases and relatives and those reported in the MARTHA control population for each CpG site probed in the HM450 array. The 'epi-Manhattan' plot generated by this analysis showed that the *MMACHC* epimutation was the single change greater than 0.3 that reached a high level of statistical significance (Fig. 3a). In the control population, the CpG island

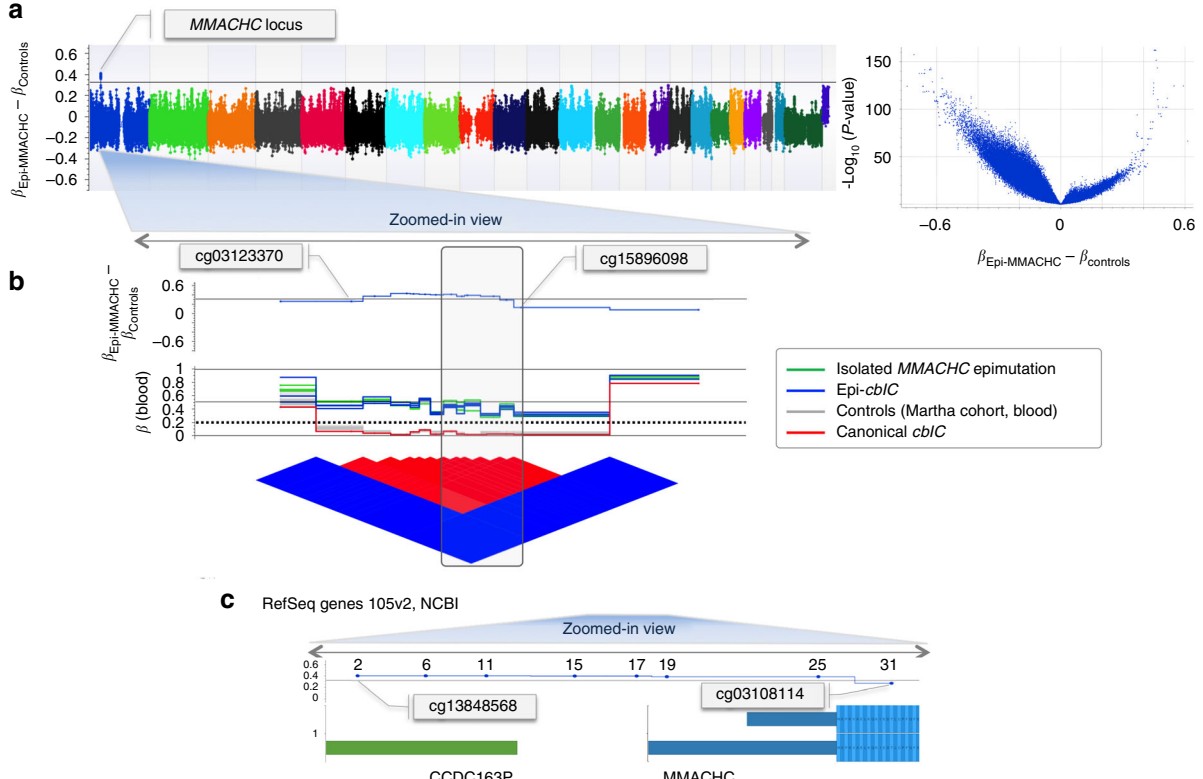

**Fig. 3** Epigenome-wide analyses confirm the presence of the epimutation in genomic DNA and sperm. **a** Epi-Manhattan plot for the differences between $\beta$ values in the epi-*MMACHC* subjects and controls ($\beta_{\text{Epi-MMACHC}}-\beta_{\text{Controls}}$). The horizontal line indicates a difference of 0.3; the volcano plot shows the relationship between ($\beta_{\text{Epi-MMACHC}}-\beta_{\text{Controls}}$) and $-\log_{10}$ (*P*-value) for the *t*-test. **b** Enlarged view of the *MMACHC* locus; $\beta$ values of the CpG probes in the *MMACHC* locus and the epi-linkage disequilibrium plot reporting the matrix relationship between the CpG probes based on the blood samples. The dotted line corresponds to a $\beta$ value threshold of 0.2, below which the CpG probe was considered to be fully unmethylated. **c** Genomic position of the *MMACHC* locus according to RefSeq Genes 105v2, NCBI (GRCh37)

that encompassed the promoter and the first exon of *MMACHC* was hypomethylated[14] and surrounded by hypermethylated CpG sites (Fig. 3b, c). This corresponded to an 'epi-haplotype block' composed of the eight CpG sites probed by the HM450K array (Fig. 3b, c). Expression of *MMACHC* is regulated by a transcription regulatory complex of proteins that involves HCFC1, THAP11, and ZNF143[15]. Analysis of the HM450K methylome failed to find any aberrant hypermethylation related to *HCFC1-ZNF143-THAP11* in our epi-*cblC* cases.

We performed ChIP-Seq analyses on the chromatin of case and control fibroblasts to further characterize the epigenetic changes at *MMACHC* and neighboring genes (Fig. 4a). We observed a significant accumulation of trimethylated lysine 36 on histone H3 (H3K36me3) in the *MMACHC* promoter region and the proximal part of the 5′ adjacent *CCDC163P* gene only in patients harboring the epimutation (Fig. 4a). In contrast, all control and patient samples show the absence of the H3K36me3 mark in the promoter region of *PRDX1*, the gene 3′ to *MMACHC* (which was associated with the absence of DNA hypermethylation).

**The *MMACHC* epimutation occurs secondary to a *PRDX1* mutation**. We performed whole-genome sequencing (WGS) of DNA from cases CHU-12122 and WG-3838 and their relatives to determine whether the promoter hypermethylation of *MMACHC* was a secondary epimutation that resulted from a genetically heritable variant[1, 2, 16]. The only sequence variant that segregated with the phenotype across all samples was in *PRDX1* (Fig. 4b and Fig. 5a). *PRDX1* is located adjacent to *MMACHC* and on the

opposite strand. We found a *c.515-1G>T* mutation of *PRDX1* in cases CHU-12122 and WG-4152 and a *c.515-2A>T* mutation of the same gene in WG-3838 (Fig. 4b). These *PRDX1* mutations were also found in the relatives who carried the secondary epimutation, i.e., the father and paternal grandfather of CHU-12122 and the father of WG-3838 (Fig. 5a). The *PRDX1* variants and the polymorphism rs3748643 associated with the secondary epimutation were present in the same allele as evidenced by DNA sequencing and transmission in the heterozygous relatives (Fig. 1c and Fig. 2b).

**Presence of the secondary epimutation in additional cases**. We replicated our results in five cases from two other centers. We found the secondary epimutation and the *c.515-1G>T PRDX1* mutation in all five cases, and confirmed the maternal transmission of the *PRDX1* mutation with the secondary epimutation in one case (Supplementary Table 1). The *c.515-1G>T* variant was present in seven individuals of European ancestry out of 66,000 individuals reported in the Exome Aggregation Consortium (ExAC) database (estimated frequency in the general population $1.052 \times 10^{-4}$).

**The epimutation results from *PRDX1* antisense transcription**. *MMACHC* belongs to a trio of reverse (R1)–forward (F2)–reverse (R3) genes with dual partial overlaps of *MMACHC* by *CCDC163P* and *PRDX1* at its 5′ and 3′ ends, respectively. The two mutations in *PRDX1* affect the canonical splice acceptor site of intron 5, 1 and 2 bp apart, respectively. They are predicted to disrupt normal

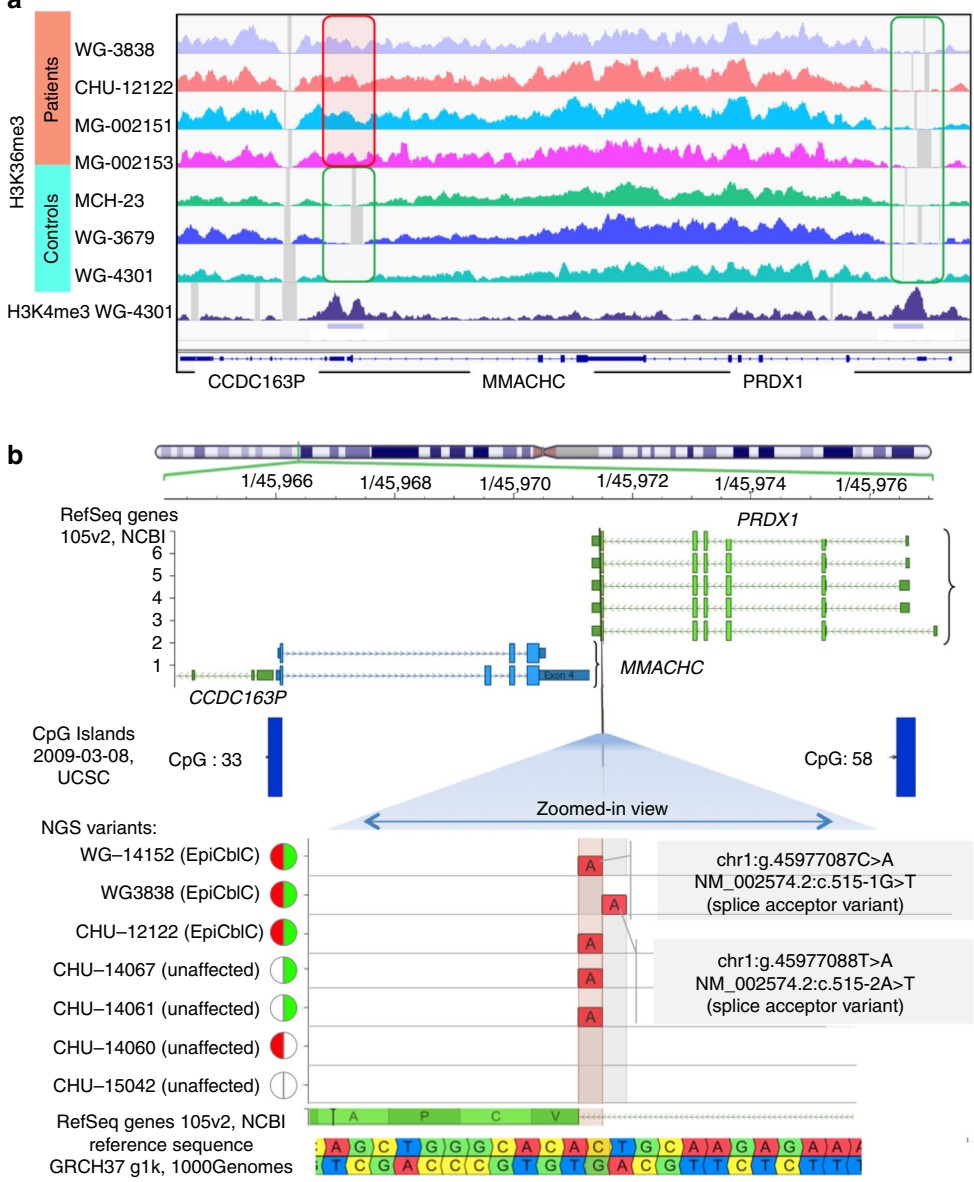

**Fig. 4** The MMACHC epimutation is associated with a H3K36me3 chromatin mark in the promoter and a mutation in the *PRDX1* adjacent gene. **a** Results of ChIP-Seq analyses at the genomic region encompassing the promoters of *MMACHC/CCDC163P* and *PRDX1*. Genomic panels show normalized coverage for histone H3 trimethylated lysine 36 (H3K36me3) mark in patients (tracks 1–4 from top) and controls (tracks 5–7). H3K4me3 track along with the peak calls, for one of the controls, has also been shown. The rectangles indicate the promoter regions. The same scale has been set in all panels. See Methods for more details on samples and analysis. **b** Identification of *PRDX1* splice acceptor variants in cases with *MMACHC* epimutation by whole-genome sequencing. Top: the vertical black line is positioned on the locus of the two *PRDX1* variants within the splice acceptor site (AG sequence) on intron 5. Bottom: zoomed view centered on the splice acceptor site of *PRDX1* intron 5. Genomic positions are reported according to the reference sequence GRCh37. The red semicircle denotes the presence of *MMACHC* genetic mutation. The green semicircle denotes the presence of *MMACHC* epimutation. The white semicircle denotes the absence of both

splicing and to produce skipping of exon 6 and the polyA transcription termination signal of *PRDX1*. We confirmed this prediction by single-strand RNA-Seq and RT-PCR analyses of case fibroblasts, and found that the loss of transcription termination of *PRDX1* produced an aberrant extension of antisense transcription through the *MMACHC* and *CCDC163P* genes. The level of antisense transcripts was high in case fibroblasts and undetectable in control fibroblasts. The activation of cryptic acceptor sites and the resulting skipping of exons and introns of *MMACHC* generated several transcripts identified in both RNA-Seq and RT-PCR. Among them, we identified a predominant 1.0 kb transcript that encompassed the exons 4 and 5 of *PRDX1*, the CpG island

and the bidirectional promoter, and the first exon of *CCDC163P* (Fig. 5b, Fig. 6a, b, c). This transcript resulted from the activation of a cryptic antisense splicing site located in the middle of *MMACHC* exon 1 (Fig. 6d, e).

We further demonstrated that the *PRDX1* mutation produced the silencing of *MMACHC* expression through the methylation of its promoter. To address this issue, we silenced the expression of *PRDX1* with siRNA transfection in WG-3838 fibroblasts and the cell line MeWo-LC1. We used WG-3838 fibroblasts because of the complete absence of detectable sense transcription of *MMACHC* exons. We showed previously that the MeWo-LC1 melanoma-derived cell line had a *cblC* phenotype and no

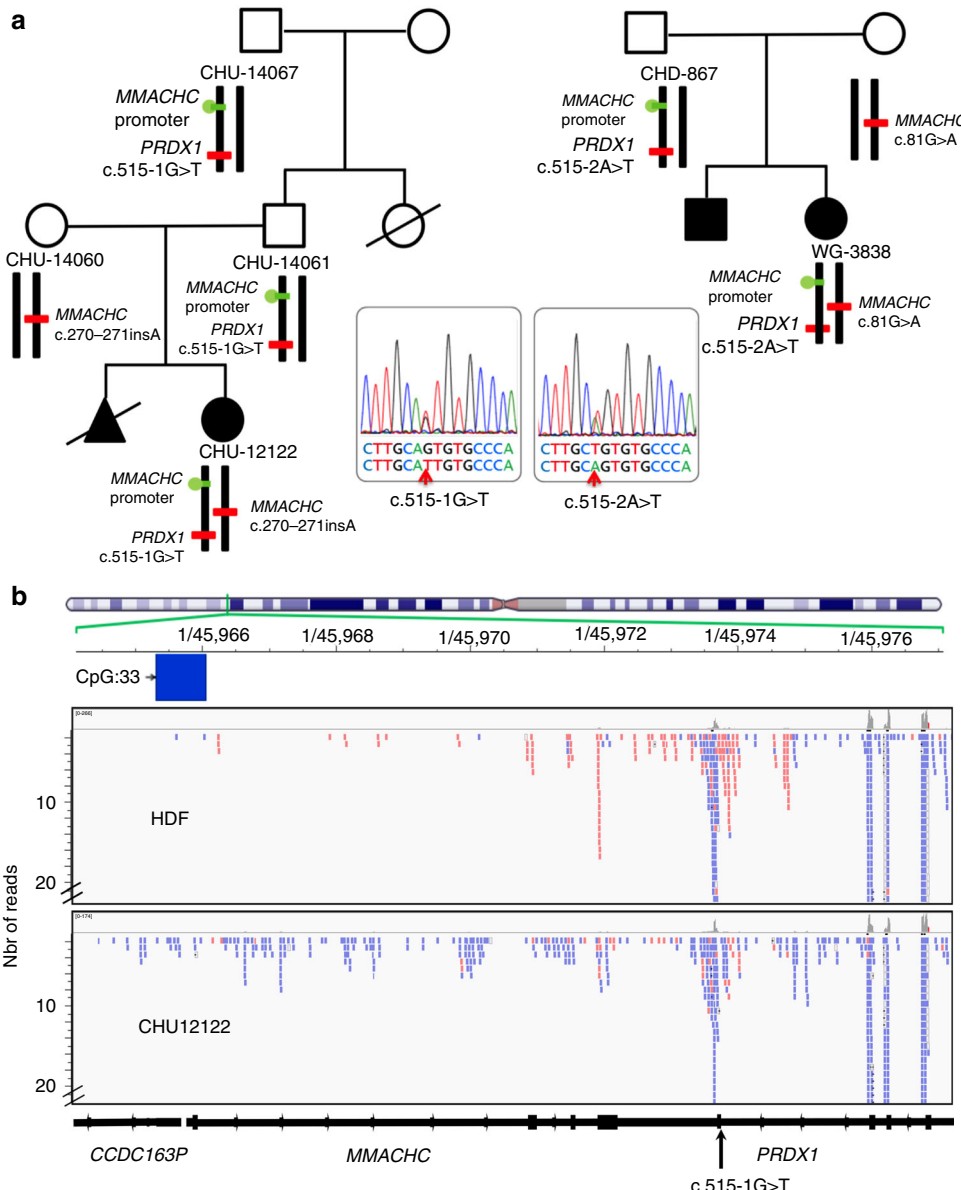

**Fig. 5** Pedigrees derived from whole-genome sequencing and epigenome analyses and RNA-Seq in case fibroblasts. **a** Pedigrees derived from WGS of DNA by HiSeq X Ten Illumina System in cases CHU-12122 and WG-3838 and their relatives. WGS evidences *c.515-1G>T* and *c.515-2A>T* mutations of *PRDX1*, in cases CHU-12122 and WG-3838 and the relatives who bear the secondary epimutation, respectively. The mutations are absent in relatives, who bear the *MMACHC* heterozygous mutation. **b** NGS sequencing of RNA from case CHU-12122 and HDF control fibroblast line. Overlapping antisense (in blue) transcription of *PRDX1/MMACHC/CCDC163P* trio of genes is predominant in CHU-12122 fibroblasts while *MMACHC* sense transcription (in red) is predominant in HDF control fibroblasts

transcription of *MMACHC* [17]. We found recently that this cell line had the same antisense transcription of *PRDX1* found in patient fibroblasts. We observed that the silencing of *PRDX1* restored the expression of MMACHC in both cell lines and that it produced 10–15% hypomethylation of the allele initially fully methylated in the clones generated from bisulfite-treated DNA. No effect was observed in control fibroblasts transfected with the *PRDX1* siRNA and in WG-3838 fibroblasts and MeWo-LC1 cells transfected with a control siRNA (Fig. 6f, g).

## Discussion

Our epi-*cblC* cases represent a cause of the autosomal recessive *cblC* disorder, resulting from a secondary epimutation in the bidirectional promoter of *MMACHC* on one allele and a coding mutation on the other. The secondary epimutation is caused by an inherited mutation of the adjacent *PRDX1* gene. It is directly involved in the mechanism of the disease through the forced antisense transcription of the adjacent mutated *PRDX1* and is present in three generations. Previous reports have described DNA methylation at genes that mimics recessive mutations by creating a transcriptional haploinsufficiency with tissue-specific epigenetic silencing[18]. In contrast, the secondary epimutation seen in our cases is present in the DNA from blood cells, fibroblasts, and sperm. The high expression of *PRDX1* in spermatic cells may explain the presence of the epimutation in sperm, in contrast to the spermatozoa erasure previously observed in families with epigenetic silencing of the *MLH1* gene[5, 14].

The secondary epimutation of our epi-*cblC* cases is located in a trio of genes (reverse *CCDC163P*–forward *MMACHC*–reverse *PRDX1*) (Fig. 6d, e), in a bidirectional promoter in the head-to-

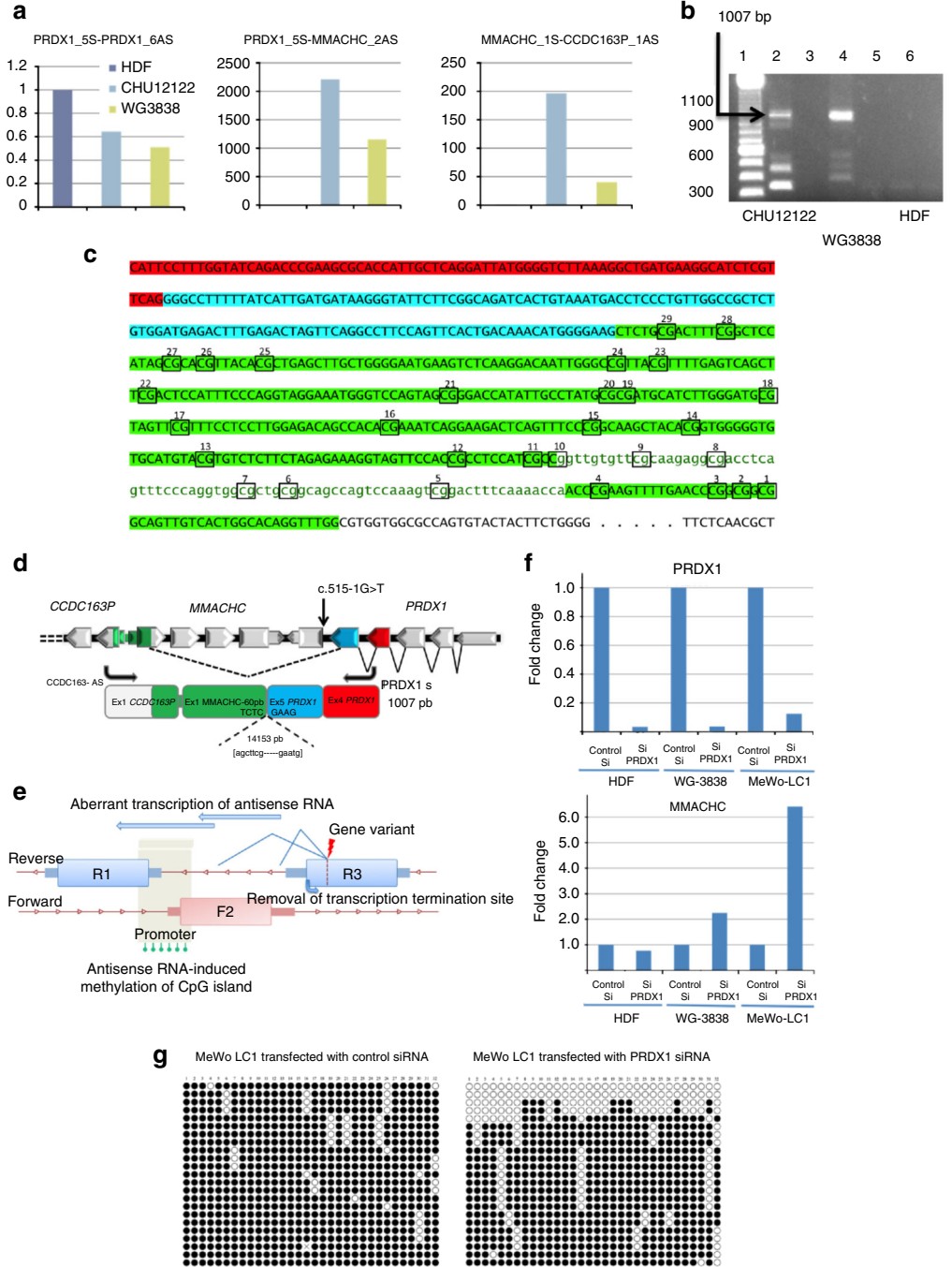

**Fig. 6** The *PRDX1* mutation causes aberrant antisense transcription through the *MMACHC* and *CCDC163P* genes. **a** qRT-PCR analyses of antisense transcripts encompassing exons 5 and 6 of *PRDX1*, *PRDX1* exon 5—*MMACHC* exon 1 and *MMACHC* exon 1-*CCDC163P* exon 1 showing the aberrant extension of *PRDX1* antisense transcription through sense *MMACHC* and antisense *CCDC163P* and no aberrant transcripts in HDF control fibroblasts. **b** RT-PCR detection of an aberrant 1 kb transcript produced with *PRDX1* sense and *CCDC163P* antisense primers. Lanes 2, 4, and 6 correspond to RT-PCR of RNA from fibroblasts CHU-12122, WG-3838, and HDF (control fibroblasts). Lanes 3 and 5 correspond to control experiments without fibroblast RNA in the reaction mixture. They show no amplification artifacts. **c, d** Sanger sequencing of cDNA shows that the antisense aberrant transcript encompasses *PRDX1* exons 4 (red) and 5 (blue), part of exon 1 and promoter of *MMACHC*, and part of exon 1 of *CCDC163P* (green). **e** Proposed mechanisms of epigenetic silencing of the non-mutated *F2* allele in patients with a heterozygous mutation of a causal gene that belong to a trio of reverse (R1)–forward (F2)–reverse (R3) genes. **f** Expression of *PRDX1* and *MMACHC* in control fibroblasts, fibroblasts of patient *WG3838*, and MeWo-L1 cell line transfected with control and *PRDX1* siRNA. **g** *MMACHC* methylation epigrams of MeWo-LC1 cells transfected with control and *PRDX1* siRNA

head sense–antisense gene pair (SAGP) constituted by *CCDC163P* and *MMACHC*. The forced antisense transcription of *MMACHC* resulted from the skipping of the last exon of *PRDX1*. This skipping was the causative defect that produced the epimutation since the silencing of *PRDX1* decreased the methylation

of exon 1 of *MMACHC* and the promoter, and restored the transcription of *MMACHC*, in WG-3838 and MeWo-LC1 cells. However, the silencing of *PRDX1* produced only 10–15% hypomethylation of the allele initially fully methylated, suggesting a limited reversibility of the epimutation (Fig. 6g). Antisense

transcription could lead to RNA polymerase collision and the formation of triplexes in the bidirectional promoter[19]. The formation of triplexes can generate CpG methylation, as shown for *PARTICLE* and *MAT2A* SAGP in cells exposed to radiation[20]. Recent data also strongly support a role of antisense transcription of SAGP in the generation of H3K36me3 marks, in the developmental mechanisms of yeast, and in the *Igf2r* promoter of mouse embryonic stem cells (ESC)[21]. The H3K36me3 mark could play a major role in the de novo methylation of the CpG island. It may reflect the rate of antisense transcription as it is deposited by the histone–lysine *N*-methyltransferase SETD2 at active genes, where it binds to the active form of RNA polymerase II[22]. The de novo methylation of CpG islands is preferentially targeted to genomic regions with elevated H3K36me3 levels through the recruitment of DNMT3B1 in mouse stem cells[23].

To our knowledge, the epi-*cblC* entity is the first example of human epigenetic disease documented by the presence of a secondary epimutation in three generations and its presence in sperm of the probands' fathers. Gene silencing by methylation through a gene disruption that caused antisense transcription across the CpG island of the promoter was previously reported for *HBA2* (alpha-thalassemia) and *MLH1* (familial cancer syndrome)[24–26]. In contrast to our data, epigenetic silencing of *HBA2* and *MLH1* caused by antisense transcription is not maintained in sperm[24–26]. In another example of familial cancer, the epigenetic somatic inactivation of the *MSH2* allele resulted from the extension of the sense transcription of the upstream *EPCAM* (formerly *TACSTD1*) gene produced by microdeletions *in cis*[27]. The presence of the *MMACHC* secondary epimutation in DNA from sperm, fibroblasts, and blood may be explained by the ubiquitously high expression of *PRDX1* in germ cells, stem cells, and somatic cells. The high expression of *PRDX1* in germ cells may explain why the secondary epimutation escaped erasure in spermatozoa, in contrast to previous reports of other diseases[5,14]. The high expression of *PRDX1* could also maintain this epimutation during early embryonic development. The transcription of *PRDX1* has been studied in bovine embryos from the two-cell to the blastocyst stages[28]. *PRDX1* transcripts were detected throughout development, including in oocytes before and after maturation and in 2c, 5–8c, 9–16c embryos, morulae, and blastocysts. *PRDX1* was also ubiquitously expressed in E7–E10 mouse embryos (http://www.informatics.jax.org; http://dbtmee.hgc.jp/) and adult humans (http://www.proteinatlas.org). The methylation of the promoter in sperm is also consistent with previous results in a mouse model, in which genes with high H3K36me3 levels showed elevated gene-body DNA methylation in sorted germ cells of pooled E13.5 testes[29].

The cases of epi-*CblC* provide the first example of a rare disease produced by compound epigenetic/genetic heterozygosity in a reverse1 (R1)–forward2 (F2)–reverse3 (R3) trio of genes (Fig. 6d, e). This suggests consideration of this mechanism in cases of recessive diseases bearing a single heterozygous variant in the causal gene in the presence of typical disease manifestations. The number of R1–F2–R3 trios of genes with a configuration similar to the *CCDC163P–MMACHC–PRDX1* trio is limited in the human genome[30]. We have listed examples of such trios with a recessive transmission of inherited diseases produced by F2 gene mutations (Supplementary Table 2). In most of them, the CpG island is hypomethylated and the RNA-Seq analysis of control fibroblasts shows the absence of aberrant transcription of the adjacent genes[31,32]. Our data suggest that the presence of an epimutation in patients with a single heterozygous mutation and a severe phenotype for any of these diseases should be systematically investigated.

In contrast to coding mutations, pharmacological inhibitors of DNMTs such as 5-AZA can reverse epigenetic changes.

Consistently, treatment of fibroblasts from one of our epi-*cblC* cases with 5-AZA restored bi-allelic *MMACHC* expression with detection of the wild-type allele (Fig. 2a). This suggests that further evaluation of the use of this compound in the treatment of potentially fatal, severe decompensation in these 'epi-diseases', as observed in our two pediatric cases, is warranted.

In conclusion, we report cases of a rare metabolic disease caused by compound heterozygosity of a secondary epimutation detected in somatic cells and sperm in one allele and a genetic mutation in the other. These cases represent an entity that we named "epi-*cblC*". This research reveals perspectives on the diagnosis and treatment of inherited metabolic diseases in general and "epi-*cblC*" patients in particular. It suggests to search for epimutations in rare diseases with typical severe presentation despite heterozygous gene mutation.

## Methods

**Patients**. Informed written, signed consent for performing the analyses was obtained from all subjects and from the parents, when appropriate. We followed the rules of the French National Reference Center of Inborn Metabolism Diseases of Nancy that were reviewed and approved by The Haute Autorité de Santé (HAS, Sain Denis, France, https://www.has-sante.fr/) for the study of the patients and their relatives. Case 1 (CHU-12122) was the first child of an unrelated Caucasian couple, with a healthy paternal half-brother and a healthy paternal half-sister. A history of Sudden Infant Death Syndrome was reported for the father's sister. The mother had three episodes of spontaneous abortion of unknown origin. The proband was born at term with a low birth weight of 2450 g. After 1 month of breastfeeding, she had marked mucocutaneous pallor, tachycardia, and axial hypotonia. Further investigations during hospitalization showed megaloblastic anemia with 3.5 g/dL hemoglobin, discrete concentric myocardial hypertrophy, pulmonary hypertension, and hepatomegaly. Serum vitamin B$_{12}$ and folate levels were within the reference ranges. Impaired activities of methionine synthase and methylmalonyl-CoA mutase were reflected by decreased methionine at 5 μmol/L (24–38) and increased homocysteine and methylmalonic acid at 174 μmol/L (reference range <0.15 μmol/L) and 325 μmol/L (reference range <0.35 μmol/L), respectively. She was diagnosed with *cblC* by fibroblast complementation analysis. The patient died at 1 month of age after multiple organ failure with hypotension, desaturation, and severe bradycardia.

Patient WG-3838 was born after an uneventful full-term pregnancy and weighed 3400 g. Her older brother died at 1 month of age from cardiac arrest, and a cardiomyopathy was noted upon autopsy. At week 5, the patient was taken to the emergency room. Serum vitamin B$_{12}$ was within the reference range, serum methionine was low, and homocysteine and MMA were increased to 110 and 20 μmol/L, respectively. The patient was hypertensive and developed tachycardia. She had a sudden cardiac arrest and developed acute renal failure. She had a second cardiac arrest and died at age 2 months. She was diagnosed as a *cblC* case by fibroblast complementation analysis.

Patient WG-4152, a 59-year-old male, had a history of squamous cell carcinoma of the left palatine tonsil. Four months after surgery, he developed an oral thrush and vomiting and underwent several tests including metabolic analyses. All results were within the reference range, except for MMA (9.0 μmol/L) and homocysteine (99 μmol/L). He was diagnosed as a *cblC* case by fibroblast complementation analysis.

**Control populations**. The Asthma BioRepository for Integrative Genomic Exploration consortium (Asthma BRIDGE consortium, USA) is an open-access collection of cDNA and DNA from more than 1450 well-characterized subjects participating in ongoing genetic studies of asthma and an accompanying database of phenotype, genome-wide SNP genotype, gene expression, and methylation data[11]. The Caucasian population of the MARTHA (MARseille THrombosis Association) cohort was recruited in the Thrombophilia Centre of La Timone University Hospital (Marseille, France) between January 1994 and October 2005[12]. The MH450K profiling data of the sperm DNA from the control population in Utah are publicly available at the Gene Expression Omnibus under #GSE64096, Utah cohort[13].

**Studies in skin fibroblasts**. We measured the cellular incorporation of the label from 1-[14C]propionate and 5-[14C]methylTHF into cellular macromolecules that precipitated in 5% trichloroacetic acid at 5 °C to evaluate the metabolic functions of methylmalonylCoA mutase (MUT) and methionine synthase (MS), respectively. If incorporation was significantly reduced, complementation analysis was performed. Patient cells were fused with fibroblasts from patients with different inborn errors of cobalamin metabolism, and the incorporation of labeled [14C]propionate and 5-[14C]methylTHF was measured in parallel in fused and unfused cells. Decreased incorporation was complemented by fibroblasts from all classes of inborn errors except for the class to which the patient belonged. These tests were used to assign

patients to known complementation classes[6]. Treatment with 5-aza-2′-deoxycytidine (5-aza-dC) was performed in fibroblasts from case CHU-12122. Fibroblasts were seeded at $3 \times 10^5$ cells/100-mm dish 24 h prior to treatment in DMEM supplemented with 10% fetal calf serum at 37 °C and 5% $CO_2$. Cells were exposed for 72 h to 10 µM 5-aza-dC (Sigma-Aldrich Chimie S.a.r.l, St. Quentin, France). The culture media was replaced every 24 h with fresh media containing 5-aza-dC. At day 3, TRIzol® reagent (Thermo Scientific™, Fontenay-sous-Bois, France) was used to extract the total RNA from the cultured fibroblasts. RT-PCR was performed with RNA from treated and non-treated cells, using the primers listed in the Supplementary Table 3. First-strand cDNA was synthesized from 2 µg of total RNA using the SuperScript First-Strand Synthesis System (Invitrogen). PCR amplification of the whole open-reading frame of *MMACHC* was performed with Phusion High-Fidelity DNA Polymerase (Thermo Scientific, Fontenay-sous-Bois, France) under standard conditions. The primers used for *MMACHC*_cDNA synthesis were: forward, 5′-CAGCAAGCTCAGCGTGTAAC-3′, reverse 5′CCAC-CATAAATCAGGGTCCA-3′. The RT-PCR amplicon was sequenced by direct sequencing using the BigDye Terminator v.3.1 kit (Thermo Scientific™, Fontenay-sous-Bois, France) and the same primers.

For RNA-Seq and RT-qPCR study of aberrant transcription, RNA depleted of rRNA transcripts was extracted from case and control fibroblasts using RiboMinus™ (Thermo Scientific, Villebon, France). cDNA libraries were prepared using the TruSeqTM RNA Sample Preparation Kit (Illumina, San Diego, CA, USA). RNA sequencing was performed using the Illumina Hi-scan sequencer. Reads were classified according to known 5′ and 3′ boundaries of annotated genes. For RT-PCR study of aberrant transcription, first-strand cDNA was synthesized from 2 µg of total RNA using random primers with SuperScript II reverse transcriptase (Invitrogen, Mantes la Jolie, France) according to the manufacturer's protocol. PCR reactions were performed with Phusion High-Fidelity DNA Polymerase (Thermo Scientific™, Fontenay-sous-Bois, France). Primers for the synthesis of PRDX1-CDC1623P 1007 bp transcript were PRDX1-S forward 5′-CATTCCTTTGGTATCAGACCCG-3′ and CCDC163P-AS reverse 5′-AGCGTTGAGAAGCACATCCA-3′. The expression of the aberrant transcript was quantified by real-time PCR using PrimeScript™ RT Master Mix and SYBR Premix Ex Taq (Takara, Ozyme St Quentin en Yvelines, France) on a StepOne Plus machine (Applied Biosystems) with Pol II as an internal control.

**DNA extraction**. Genomic DNA was extracted from confluent cultures of fibroblasts using the QIAmp kit (Qiagen, Courtaboeuf, France). Genomic DNA from EDTA-treated peripheral blood samples was isolated using the *Nucleon BACC3 Kit*. (GE Healthcare, Aulnay-sous-Bois, France). Sperm DNA was extracted from spermatozoa isolated by a swim-up protocol to discard contaminating somatic cells. Microscope examination confirmed the purity of spermatozoa and absence of somatic cells. Furthermore, the absence of somatic cells was proven by methylation analysis of SNRPN, a robustly imprinted gene. All SNRPN reads are unmethylated in the DNA extracted from our sperm preparations. The sperm sample was heated at 56 °C for 2 h with frequent shaking in the lysis buffer, and DNA was extracted using the DNeasy Blood and Tissue Kit (Qiagen, Courtaboeuf, France).

**Determination of DNA methylation**. The methylation status of the CpG island in the 5′-region of the *MMACHC* gene (RefSeq: Chr1:45,965,587 to Chr1:45,966,049) was determined by bisulfite conversion, cloning, and sequencing of individual clones. Six-hundred nanograms of genomic DNA was converted by bisulfite using the EZ DNA Methylation-Gold kit (Zymo Research, Proteigene, Saint-Marcel, France). The bisulfite-treated DNA was amplified by PCR (primers: forward 5′-TTAAATTTGTGTTAGTGATAATTGT-3′, reverse 5′-AACTAACC-TAAAAAAAATAAACCTC-3′) using ZymoTaq DNA Polymerase (Zymo Research). The amplicon was inserted into the pCR4-TOPO vector (Invitrogen, Life Technologies, Courtaboeuf, France), and individual clones were sequenced in both directions using universal M13 primers and the BigDye Terminator v.3.1 Sequencing Kit (Applied Biosystems, Courtaboeuf, France). Only DNA strands that were >95% converted were used for analysis. Methylome analysis for genome-wide profiling of bisulfite-converted DNA was determined using the Infinium HumanMethylation450 BeadChip array (Illumina, Paris, France), according to the manufacturer's instructions. Probe annotation information including sequence and chromosome location for the Infinium HumanMethylation450 BeadChip array was retrieved from the HumanMethylation450 v1.2 manifest file.

**Gene sequencing**. WGS was performed using our standard protocols at the McGill University and Génome Québec Innovation Center[33]. All samples were sequenced using the HiSeq X Ten Illumina System sequencer with paired-end 150-bp read length and the mean average of 35–37X. Bioinformatics analysis was performed as previously described[34]. Briefly, the Burrows-Wheeler Aligner (BWA-MEM, version 0.7.7) was used to map all reads to UCSC hg19. PCR duplicates were removed from alignments using Picard version 1.96 (http://picard.sourceforge.net). Indels were realigned using the Genome Analysis Toolkit (GATK). SNVs and short indels were called using GATK haplotype caller (version 3.4–46). All variants were annotated with ANNOVAR and in-house scripts, and most likely protein damage variants (nonsense, splice site, frameshift indel, and missense) were considered for further analysis. The variants were also annotated for the allele frequency in public

databases such as ExAC (http://exac.broadinstitute.org/), the National Heart, Lung, and Blood Institute (NHLBI) exomes (V.0.0.14, http://evs.gs.washington.edy/EVS/), the 1000 Genomes Project database, and our in-house database of >2000 exomes. In order to remove common variants and sequencing artifacts, variants with MAF >0.001 in any of the aforementioned databases were removed from further analysis.

**Whole-genome ChIP-Seq**. ChIP-Seq was performed in patient's fibroblasts and control fibroblast lines without inherited metabolic defects in the cobalamin pathway. MCH23 fibroblasts were obtained from a healthy 3-year-old male, WG-4301 fibroblasts were from a 3-month-old female, and WG-3679 fibroblasts were from a 5-year-old female with *mut* mutation. For crosslinking, cells were directly fixed with formaldehyde (final concentration of 1%) for 10 min with gentle shaking at room temperature. The fixed cells were then quenched with glycine at a final concentration of 138 mM for 5 min at room temperature. The crosslinked cells were subsequently washed with 10 mL ice-cold PBS and scraped. The cells were pelleted by centrifugation at 5000×$g$ for 5 min at 4°C. For cell and nuclear lysis, the crosslinked cell pellets were resuspended in ice-cold Cell-Lysis buffer (5 mM PIPES pH 8.5, 85 mM KCl, 1% IGEPAL CA-630, 50 mM NaF, 1 mM PMSF, 1 mM phenylarsine oxide, 5 mM sodium orthovanadate, 1× Roche complete mini EDTA-free protease inhibitors) at 500 µL/10 million cells and incubated for 15 min on ice. The nuclei were pelleted by centrifugation at 5000×$g$ for 10 min at 4°C. Nuclei were lysed in Nuclear-Lysis buffer (50 mM tris-HCL pH 8.0, 10 mM EDTA, 1% SDS, 50 mM NaF, 1 mM PMSF, 1 mM phenylarsine oxide, 5 mM sodium orthovanadate, 1× Roche complete mini EDTA-free protease inhibitors) at 100 µL/1 million cells adjusting by 50 µL for every excess of 10 million additional cells. The nuclei were incubated for 30 min on ice and then transferred to 1.5 mL TPX tubes (Diagenode) at 100–250 µL aliquots. The lysed nuclei were sonicated with a UCD-300 Diagenode Bioruptor on high power at 4°C with repeated cycles. The chromatin size distribution was evaluated on 2% pre-cast E-gels (Invitrogen). Samples with chromatin size distribution >500 bp were further sonicated. For chromatin concentration and pre-clearing, the material was first spun down at 12,000×$g$ for 10 min at 4°C. The resulting pellets were washed with 10 mM tris-HCl pH 8.0 and pooled with collected supernatant for each sample. The chromatin material was then concentrated to a desired volume equivalent to 50 µL/1 million cells and <0.1% SDS using Nanosep 10k omega centrifugal devices (PALL life sciences). The chromatin was then pre-cleared with 50 µL Protein A Dynabeads for 1 h at 4°C with rotation. The Immunoprecipitation and reverse crosslinking were performed with the IP-Star SX-8G (Diagenode) using the Auto-Histone ChIP kit (Diagenode) according to the manufacturer's instructions. Input material was prepared by following the reverse crosslinking procedure above, but without immunoprecipitation. The eluted material was collected by placing the samples on a magnetic rack. DNA was extracted using Qiagen's MinElute PCR purification columns. ChIP-Seq libraries were prepared using the Kappa Library Preparation Kit Illumina Platforms (Kappa Biosystems) and Illumina TruSeq adapters (Illumina) according to the manufacturer's instructions. DNA clean-up and size selection was performed using Ampure XP Beads. Samples were then sequenced using the HiSeq 2000 with selected read length of 50 bp. Sequence reads were aligned as explained for the WGS bioinformatic analysis. Duplicated reads were removed using PICARD. Peaks were called using Model-based Analysis of ChIP-Seq MACS2 with input DNA as control and using the broad peak mode (--broad --broad-cutoff 0.1)[35]. In order to generate Wiggle (WIG) and Tiled Data Format (TDF) files, HOMER (4.7)[36] and igvtools (2.3.67; http://software.broadinstitute.org/software/igv/igvtools) were used. Tags were normalized to 10 million reads to generate tracks for visualization.

**Effect of *PRDX1* silencing on *MMACHC* methylation**. The human melanoma cell line MeWo-LC1 derives from MeWo cells (ATCC® HTB-65™). It was obtained from Dr. Robert Liteplo (University of Ottawa, Ottawa, Ontario). The human melanoma cell line MeWo-LC1 variant of MeWo cells, fibroblasts from case WG-3838, and fibroblasts HDF for control were cultured in DMEM medium supplemented with 4.5 g/L glucose, 10% v/v heat-inactivated fetal bovine serum, 1% v/v pyruvate in a humidified atmosphere with 5% $CO_2$ at 37 °C. The cell lines were tested for absence of mycoplasma contamination. The cells were seeded into six-well plates at a concentration of $0.5 \times 10^5$ cells/well the day before transfection; 10 nM of siRNA against *PRDX1* gene (GGUCAAUACACCUAAGAAA, GGGCA-GAAGAAUUUAAGAA, GAUGGUCAGUUUAAAGAUA, and CUU-CAAAGCCACAGCUGUU) was transfected with Lipofectamine RNAiMAX transfection reagent (Invitrogen) into MeWo-LC1 for 72 h at 37 °C. Non-silencing siRNA (AllStars Negative Controls—Qiagen) was used as a negative control. After a 72-h period of incubation, the cells were collected and used for mRNA and DNA isolation. Two-hundred nanograms of RNA was reverse-transcribed in a 10-µL reaction volume using PrimeScript™ RT Master Mix (TAKARA Bio, USA) according to the manufacturer's recommendations; 2 µL of cDNA was used for qPCR with SYBR® Premix Ex Taq™ (Tli RNaseH Plus) (TAKARA Bio, USA) in a 20-µL reaction volume according to the manufacturer's instructions, with reverse and forward primers at a concentration of 0.2 µM. Specific amplifications were performed using the following primers: *MMACHC* forward 5′-ATCTGGGCCGTGTTAGAGAGA-3′, reverse 5′-CCTCCA-CATCTTGTCGTTGG-3′; *PRDX1* forward 5′-CCACGGAGATCATTGCTTTCA-3′, reverse 5′-AGGTGTATTGACCCATGCTAGAT-3′. Quantification was

performed using the housekeeping gene RNA polymerase II as an internal standard with the following primers: forward 5′-CAGACCGGCTATAAGGTGGA-3′, reverse 5′-GGTAGACCATGGGAGAATGC-3′. Temperature cycling for *MMACHC* and *PRDX1* was 30 s at 95 °C followed by 40 cycles consisting of 95 °C for 5 s and 61 °C for 30 s. The PCR program for RNA polymerase II was 30 s at 95 °C followed by 40 cycles consisting of 95 °C for 5 s and 60 °C for 30 s. Results were expressed as arbitrary units by calculating the ratio of crossing points of amplification curves of *MMACHC* or *PRDX1* and the internal standard by using the δδCt method. The bisulfite-treated DNA from MeWo-LC1 was used for methylation analysis of the *MMACHC* gene promoter by PCR amplification, cloning, and Sanger sequencing.

**List of sense–antisense trios of genes**. We established a list of tail-to-tail SAGPs using a systematic search through the United Sense-Antisense Gene Pairs Database and retrieved 385 gene pairs[30]. Each gene pair was visually inspected (RefSeq Genes 105v2) in order to look for the presence of a flanking gene that meets the two criteria of sense–antisense trios of genes: (1) antisense with one of the tail-to-tail SAGPs and (2) sharing a CpG island with the proxy gene (CpG Islands 2009–03–08, UCSC)[30, 31]. Among the 385 manually inspected gene pairs, 99 had a triplet gene configuration according to a reverse(R1)–forward(F2)–reverse(R3) pattern ($n = 82$) or a forward(F1)–reverse(R2)–forward(F1) pattern ($n = 17$), with a CpG island in the promoter of R1/F2 and R2/F1, respectively.

**Statistical analyses**. The comparison of CpG beta values between the cases and controls was carried out using a *t*-test, and multiple testing corrections were performed using the Bonferroni adjustment. To assess the methylome architecture (epi-haplotype blocks, epi-LD analysis), linkage disequilibrium (LD) pairwise analysis was performed on all adjacent CpG pairs in a chromosome or within a haplotype block using a matrix output for both the expectation–maximization (EM) algorithm and the composite haplotype method (CHM)[37]. $R^2$ values were used in the epi-LD plots. All epi-LD analyses were performed after transforming the CpG beta values to categorical variables according to the ENCODE project[10]. Any beta value ⩾0.6 was considered fully methylated (MM). Any beta value ⩽0.2 was considered to correspond to a fully unmethylated DNA. Beta values strictly greater than 0.2 and strictly less than 0.6 were considered to correspond to a partially methylated DNA (one of the two alleles). To assess the population stratification according to their whole methylome profile, numeric principal component analysis was performed using normalized beta values of each CpG probe across the Infinium HumanMethylation450 BeadChip array[38, 39]. The top 10 principal components (eigenvectors) were calculated with their respective eigenvalue (EV). Population stratification was assessed both in 2-dimensional and 3-dimensional visualization based on the eigenvalues of the top two and three principal components, respectively. All analyses were performed using the SNP & Variation Suite (SVS) 8.4.2 (Golden Helix, Inc. Bozeman, MT, USA).

**Data availability**. The RNA-Seq data that support the findings of this study have been deposited in the European Genome-phenome Archive (https://www.ebi.ac.uk/ega/datasets) with the accession code EGAD00001003707. The other data supporting the findings of this study are available from the corresponding author upon reasonable request.

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

## Acknowledgements

Institutional grants were received from the Region Lorraine, i-SITE Lorraine University of Excellence (LUE) and the French National Institute of Health and Medical Research (Inserm). Funding was also received from the Canadian Institutes for Health Research. We would like to thank the mother, father, and relatives of case CHU-12122 for their

valuable contribution and help in the collection of data and samples used to study the entire family, and the scientists of the McGill University and Genome Quebec and the Genomic Platform of the FR3209 CNRS-Inserm, University of Lorraine, for expert advice and performing high throughput sequencing. W.K.C received support from the Simons Foundation and the JPB Foundation.

## Author contributions

J.L.G. recruited case CHU-12122 and her relatives, designed and coordinated the study, interpreted the data, and wrote the paper. C.C. participated in the study design, coordinated and performed experiments, interpreted the data, and participated in writing the paper. J.L.G. and C.C. shared equal contribution. A.O. participated in the design of the study, performed bio-info analyses, interpreted the data, and participated in writing the paper. D.S.R. and D.W. coordinated the metabolic evaluation of WG-3838 and WG-4152 cases, participated in the study design, interpreted the data, and participated in writing the paper. J.M. and J.N. performed analyses of WGS, RNA-Seq and genome-wide ChIP-Seq, interpreted the data, and participated in writing the paper. W.K.C., J.F.B., M.B., and C.C. provided clinical and metabolic data and DNA and cells of the replication study, analyzed and interpreted the related data, and revised the manuscript. M.P. and A.B. performed cell experiments, analyzed and interpreted the related data, and revised the manuscript. T.J., J.F., A.R., and S.F. performed experiments and analyzed and interpreted the related data. A.M. performed bioinformatic analyses of WGS and genome-wide ChIP-Seq. T.P. and F.H. helped in designing and carried out the ChIP-sequencing experiments. I.M. and V.M. performed RNA-Seq analyses. D.T., B.R., and P.E.M. provided the control populations, analyzed and interpreted the related data, and participated in the preparation of the paper. D.C. interpreted the data and participated in the preparation of the paper. F.F., C.B., I.K., and I.G. performed clinical and metabolic reporting and sample collection, interpreted the data, and revised the manuscript. All authors discussed the results and commented on the manuscript.

## Additional information

**Competing interests:** The authors declare no competing financial interests.

