## [Peer Review File · Nature Communications]

Reviewer #1 (Remarks to the Author):

The authors present several cases with a severe clinical diagnosis of an inborn error of cobalamin metabolism, which would usually be caused by recessive mutations of the MMACHC gene, who instead have a compound genetic coding mutation of one allele of MMACHC and an epimutation of the other allele.

The data presented are compelling and, in my opinion, their conclusions are consistent with their data. Firstly, they demonstrate through bisulphite sequencing flanking an (uninvolved) promoter SNP in at least two of their cases that the MMACHC promoter methylation was monoallelic. They also show through allelic expression analyses of cDNA that this resulted in allelic loss of expression of one allele, and the allele affected by the epimutation was the wild-type allele (hence in trans with the germline coding mutation). They demonstrated these molecular characteristics in samples from other members of the respective families from prior generations, including the blood and, interestingly, also in the sperm of the carrier fathers. Hence they provide strong evidence that the MMACHC epimutations were transmitted from one generation to the next, intact via the gametes. This is the first time, to my knowledge, that this has been demonstrated in humans. In other examples of epimutations, the epigenetic manifestations are erased in the spermatozoa, even if the epimutation is subsequently reinstated in the somatic cells of the offspring. These findings are also consistent with what, in the field, we term a "secondary" epimutation (follows Mendelian laws of intergenerational transmission because it is caused by an underlying genetic mutation in cis).

This paper is also comprehensive in that, not only does it demonstrate the epigenetic manifestation provides a new mechanism of causation of this disease, but they also unravel the underlying cis-genetic cause, and furthermore, provide data from functional studies that corroborate their genetic findings. They identify the genetic basis of the epimutation as a cis-acting splice mutation of the neighboring PRDX1 gene, which is expressed in the antisense direction on the opposite strand. This genetic mutation, they demonstrate, causes exon-skipping resulting in loss of the polyadenylation (transcription termination) signal, with consequent aberrant continuation of expression of PRDX1 into the MMACHC. Their finding of excess H3K36 trimethylation through this region is consistent with this extended antisense transcription. (No other gene mutations were identified by WGS or panel exome sequencing of other metabolic genes segregating with the phenotype in all families). The authors then proceed to demonstrate via siRNA transfection that reduced PRDX1 expression partially ameliorates the methylation in appropriate cell lines. The finding of this PRDX1 genetic mutation and its functional impact are consistent with the inheritance pattern through 3 generations, the sequence of events leading to the epigenetic manifestations including elevated H3K36 trimethylation and MMACHC promoter methylation, and that the epimutation is isolated at MMACHC (which they provide convincing evidence of through their Human Methylation 450k array data analyses with relevant controls).

All in all, the findings presented are novel, in the following respects:

They show epimutation can serve as a cause for recessive disease by affecting one allele, while a genetic mutation accounts for the loss of function of the other allele.

They show the epimutation is transmitted via the gamete, likely due to the high levels of expression of the causative mechanisms via PRDX1 antisense transcription in this cell type.

I have only a few minor suggestions for amendment of the text, as follows:

1. The statement regarding the absence of promoter methylation at MLH1 in spermatozoa in cases of MLH1 epimutation would most appropriately be referenced by Hitchins et al., Cancer Cell 2011, since this 3 generation family was also linked to a particular cis-genetic variant as its likely cause. This paper demonstrated transmission of the epimutation through 3 generations on the same allele, despite the erasure of methylation in sperm.
2. They should also cite the story of EPCAM (formerly TACSTD1) causing tissue-specific epimutation of MSH2, since this also involved loss of the polyadenylation signal in the neighboring gene and consequent extension of expression into MSH2. Although this occurred in the sense direction, the similarities are notable and worthy of citation. I suggest Ligtenberg et al., Nature Genetics 2006.
3. In the Introduction, add one extra sentence to state definitively what the usual causes of this disorder are - homozygous mutations in consanguinous families? Compound heterozygous mutations? I had to re-read this section to find the word "recessive" to put the story into context.
4. The text and figures are quite dense and technically detailed, which could make this paper less accessible/readable for those without a firm knowledge of molecular genetics and epigenetics. If word count permits, perhaps a summary statement at the beginning and/or end of each section in Results to explain what the goal of the experiment is, and what it then reveals, would help. Also, a final more simplistic figure of the epimutation, its mechanistic basis, and its impact, to summarize the findings might make the take-home message more digestible.

It has been a pleasure reading this manuscript and I must congratulate the authors on a very impressive body of work.

Megan P. Hitchins

Reviewer #2 (Remarks to the Author):

The authors identify a rare genetic variant located in the PRDX1 gene that is recessive for a class of disorders of vitamin B12 metabolism. They demonstrate that this mutation affects a splice acceptor site, the consequence of which is read-through transcription of PRDX1 into the MMACHC gene in an antisense orientation. Antisense transcription through the promoter of MMACHC is associated with transcriptional silencing of this promoter and hypermethylation of the promoter

associated CpG island. There is also an increase in H3K36me3 around this site in individuals with this mutation. H3K36me3 is a mark that is associated with active transcription of PolII and is involved in splicing, supporting the other evidence of antisense read-through of the promoter of the neighbouring MMACHC (i.e. aberrant transcripts). They provide some functional evidence for this hypothesis by showing that knock down of the aberrant PRDX1 restores transcription of MMACHC. This mutation was discovered as when inherited as a compound heterozygote with loss of function mutations within the MMACHC coding region, it is sufficient to cause cblC metabolic disorder due to loss of MMACHC expression. They argue that this is an example of a constitutive epimutation that is transgenerationally inherited as the hypermethylation phenotype is present in sperm in cases with paternal transmission.

Novelty: These findings are novel in that this is a rare and novel mutation that is associated with an inherited disorder and has diagnostic value for cases of suspected cblC in which homozygous MMACHC mutations are absent. With regards to the claim of a transgenerationally inherited epimutation; I would apply caution. This is clearly a genetic mutation that links to phenotype through altering the epigenetic state, as they show. The presence of promoter hypermethylation in sperm does not demonstrate transgenerational epigenetic inheritance, (i.e. failure to be erased either in primordial germ cells and/or the post-fertilisation embryo). The evidence they present suggests that hypermethylation will be established in any tissue in which PRDX1 is expressed (including the male germ-line during spermatogenesis) as an obligatory consequence of the genetic mutation. They provide no direct evidence to suggest that there is failure to erase this during the developmental periods of genome-wide DNA methylation erasure. Furthermore, this is not the first example of read-through antisense transcription altering the epigenetic state of a promoter. Therefore, with regards to mechanistic insight, this is not novel (e.g. Tufarelli, C. et al. Transcription of antisense RNA leading to gene silencing and methylation as a novel cause of human genetic disease. *Nature Genet.* 34, 157–165 (2003).)

I think the conclusions and work presented in the manuscript are reasonable. I have some minor comments with regards to specific analyses and presentation, provided below.

I believe the greatest relevance and interest in this work will be for the diagnostic benefit of this specific disorder, rather than an insight into novel biological mechanisms.

Specific comments:

I found the manuscript difficult to read and perhaps some effort could be used to simplify language where possible, e.g. lines 279-281 “rare disease produced by compound epigenetic/genetic heterozygosity in a reverse1 (R1)/forward 2(F2)/reverse3(R3) trio of genes),”

-with the WGS, did you consider trans variants as well, if not, what do you consider a cis variant?

-the layout of the figures is confusing. E.g. Figure 1 relates to one pedigree. Figure 2a also relates to this pedigree as does the top part of 2b, whereas the middle of 2b is the next pedigree and the bottom the other patient, before 2c goes on to relate to the second pedigree. Having figures correspond to the order they are mentioned in the text AND grouped into related information would make it easier for the reader.

-Figure 6b; they use the wrong control

Reviewer #3 (Remarks to the Author):

Methylmalonic aciduria and homocystinuria, cblC type, is rare recessive disease caused by biallelic loss of function of the MMACHC gene on chromosome 1. The authors initially identified three patients in which one MMACHC allele has a DNA sequence mutation and the other one an epimutation (promoter hypermethylation). In two families the epimutation is also present in the patient's fathers, and in at least one of them, it also appears to be present in sperm. Interestingly, the epimutation is in phase with a DNA sequence mutation in a neighbouring gene (PRDX1), which is transcribed towards the MMACHC gene from the other strand (convergent promoters). The authors show that the mutation leads to skipping of the last PRDX1 exon and the transcription termination signal, so that the polymerase reads – in antisense - through the MMACH1 gene into the neighbouring gene on the other side (CCDC163P). It is likely that PRDX1 read through transcription induces methylation of the MMACHC promoter. This notion is supported by the presence of H3K36me3 at the MMACHC promoter and by hypomethylation and reactivation of MMACHC expression after silencing of PRDX1. In a consecutive study, the authors identified five more cases with an PRDX1 mutation and MMACHC epimutation.

In summary, Gueant et al. have identified an epimutation that is caused by a cis-acting DNA mutation. Similar cases involving read through transcription induced DNA methylation have been described before (e.g. HBA2 and MSH2), although in these cases aberrant methylation was not present in sperm. The authors claim that their study presents the first evidence for transgenerational transmission of an epimutation in humans.

For transparency I would like to state that I reviewed the manuscript previously for another journal. While the findings are very interesting and the paper has improved since the first submission, I have several points of criticism.

1. Aberrant HBA2 and MSH2 methylation in the afore-mentioned cases is restricted to somatic tissues, because the neighbouring genes LUC7L and EPCAM, respectively, are expressed in a tissue-specific manner. In contrast, PRDX1 is expressed in all tissues including germ cells and stem cells. Germ cell expression of PRDX1 most probably explains why MMACHC methylation also occurs in sperm. Thus, we are dealing with the transgenerational transmission of a DNA (PRDX1) mutation that causes MMACH1 methylation in each cell, and not with the transgenerational transmission of an epimutation. The MMACHC epimutation per se is not stable, because it easily lost after blocking PRDX1 read through (Fig. 6; this also needs to be discussed more thoroughly). The term "transgenerational inheritance of an epimutation" is reserved for the inheritance of a (stable) epimutation that has occurred spontaneously in the absence of any DNA sequence change during the life of a parent. It is important to distinguish between these two different modes of inheritance, especially with respect to the discussion of the inheritance of acquired traits. Thus, the authors should not use the term "transgenerational inheritance of an epimutation" for their cases.

2. The authors claim that the epimutation is also present in sperm from the fathers of CHU-12122 (CHU-14061) and WG-3838 (CDH-867). However, MMACHC methylation and SNRPN methylation (contamination control) is only shown for CHU-14061. The MMACHC and SNRPN data for CDH-867 need to be shown, also.

3. I am not convinced that the patients need to be reclassified as having a new type of Cbl inherited disorder, epi-cblC. I assume that MMACHC methylation mimics an MMACHC mutation and that the clinical features are the same.

4. Bisulfite sequencing of the MMACHC promoter in sperm DNA of 14061 should be shown in Fig 1, not Fig. 2.

5. Terminology is sometimes wrong or imprecise. Examples for wrong terminology:

Line 113 and ff: "c.-302G genotype of the rs3748643 ...2". This should read "allele" rather than "genotype". Lines 212-213 "The level of anti-sense transcripts was high in control and case fibroblasts". In the controls, there is no MMACHC antisense-transcription, just the sense PRDX1 transcription. Lines 217-219: "The activation of cryptic acceptor sites and resulting skipping of exons and introns of MMACHC ...". The exons and introns are not skipped; they are not recognized, because they are on the other strand. Line 235: "compound heterozygosity for a constitutional epimutation ... in cis and MMACC coding mutation in trans." This is an inappropriate use of the words "cis" and "trans".

Reviewer #1:

General comments: «*The authors present several cases with a severe clinical diagnosis of an inborn error of cobalamin metabolism, which would usually be caused by recessive mutations of the MMACHC gene, who instead have a compound genetic coding mutation of one allele of MMACHC and an epimutation of the other allele. The data presented are compelling and, in my opinion, their conclusions are consistent with their data. Firstly, they demonstrate through bisulphite sequencing flanking an (uninvolved) promoter SNP in at least two of their cases that the MMACHC promoter methylation was monoallelic. They also show through allelic expression analyses of cDNA that this resulted in allelic loss of expression of one allele, and the allele affected by the epimutation was the wild-type allele (hence in trans with the germline coding mutation). They demonstrated these molecular characteristics in samples from other members of the respective families from prior generations, including the blood and, interestingly, also in the sperm of the carrier fathers. Hence they provide strong evidence that the MMACHC epimutations were transmitted from one generation to the next, intact via the gametes. This is the first time, to my knowledge, that this has been demonstrated in humans. In other examples of epimutations, the epigenetic manifestations are erased in the spermatozoa, even if the epimutation is subsequently reinstated in the somatic cells of the offspring. These findings are also consistent with what, in the field, we term a "secondary" epimutation (follows Mendelian laws of intergenerational transmission because it is caused by an underlying genetic mutation in cis).*

This paper is also comprehensive in that, not only does it demonstrate the epigenetic manifestation provides a new mechanism of causation of this disease, but they also unravel the underlying cis-genetic cause, and furthermore, provide data from functional studies that corroborate their genetic findings. They identify the genetic basis of the epimutation as a cis-acting splice mutation of the neighboring PRDX1 gene, which is expressed in the antisense direction on the opposite strand. This genetic mutation, they demonstrate, causes exon-skipping resulting in loss of the polyadenylation (transcription termination) signal, with consequent aberrant continuation of expression of PRDX1 into the MMACHC. Their finding of excess H3K36 trimethylation through this region is consistent with this extended antisense transcription. (No other gene mutations were identified by WGS or panel exome sequencing of other metabolic genes segregating with the phenotype in all families). The authors then proceed to demonstrate via siRNA transfection that reduced PRDX1 expression partially ameliorates the methylation in appropriate cell lines. The finding of this PRDX1 genetic mutation and its functional impact are

consistent with the inheritance pattern through 3 generations, the sequence of events leading to the epigenetic manifestations including elevated H3K36 trimethylation and MMACHC promoter methylation, and that the epimutation is isolated at MMACHC (which they provide convincing evidence of through their Human Methylation 450k array data analyses with relevant controls).

All in all, the findings presented are novel, in the following respects:

They show epimutation can serve as a cause for recessive disease by affecting one allele, while a genetic mutation accounts for the loss of function of the other allele.

They show the epimutation is transmitted via the gamete, likely due to the high levels of expression of the causative mechanisms via PRDX1 antisense transcription in this cell type.»:

We thank the reviewer for these appreciations of our findings.

Specific remarks

Point 1: « *The statement regarding the absence of promoter methylation at MLH1 in spermatozoa in cases of MLH1 epimutation would most appropriately be referenced by Hitchins et al., Cancer Cell 2011, since this 3 generation family was also linked to a particular cis-genetic variant as its likely cause. This paper demonstrated transmission of the epimutation through 3 generations on the same allele, despite the erasure of methylation in sperm.* »

We have changed the former reference 5 (Hitchins et al, NEJM, 2007) by Hitchins et al., Cancer Cell 2011 and we have revised the sentence as follows:

“For example, an epimutation reported in 3 generations with a familial cancer syndrome, caused by epigenetic silencing of the *MLH1* gene, is erased in spermatozoa, but reinstated in the somatic cells of the next generation.”

Point 2: « *They should also cite the story of another example EPCAM (formerly TACSTD1) causing tissue-specific epimutation of MSH2, since this also involved loss of the polyadenylation signal in the neighboring gene and consequent extension of expression into MSH2. Although this occurred in the sense direction, the similarities are notable and worthy of citation. I suggest Ligtenberg et al., Nature Genetics 2006. This also involved loss of the polyadenylation signal in the neighboring gene and consequent extension of expression into MSH2.* »

We have added the following sentence in the third paragraph of discussion « In another example of familial cancer, the epigenetic somatic inactivation of the *MSH2* allele resulted from the extension of the sense transcription of the upstream *EPCAM* (formerly *TACSTD1*) gene produced by microdeletions in cis. » and we have cited the reference « Ligtenberg M.J., et al. Nat Genet 41:112-7 (2009). » (Ref 27 of the revised manuscript).

Point 3: « *In the Introduction, add one extra sentence to state definitively what the usual causes of this disorder are - homozygous mutations in consanguinous families? Compound heterozygous mutations? I had to re-read this section to find the word "recessive" to put the story into context.* »

We have revised the second sentence of the second paragraph of the introduction to make this point clearer, indicating that *cb1C* defects are usually caused by homozygous mutations or compound heterozygous mutations:

« The cases were classified as belonging to the autosomal recessive *cb1C* class of inborn errors of vitamin B₁₂ (cobalamin, Cbl) metabolism usually caused by homozygous or compound heterozygous mutations in the *MMACHC* gene.»

Point 4: « *The text and figures are quite dense and technically detailed, which could make this paper less accessible/readable for those without a firm knowledge of molecular genetics and epigenetics. If word count permits, perhaps a summary statement at the beginning and/or end of each section in Results to explain what the goal of the experiment is, and what it then reveals, would help. Also, a final more simplistic figure of the epimutation, it's mechanistic basis, and its impact, to summarize the findings might make the take-home message more digestible.* »

We have revised the language throughout the manuscript and we have reworded the headings of the paragraphs of results section (limited to 60 characters) to address the suggestion of the reviewer. We have simplified the mechanistic scheme of Fig 6e as suggested.

Point 5: « *It has been a pleasure reading this manuscript and I must congratulate the authors on a very impressive body of work.* »

We warmly thank Reviewer 1 for this very positive feedback and constructive suggestions.

Reviewer #2:

General comments: “*The authors identify a rare genetic variant located in the PRDX1 gene that is recessive for cblC class of disorders of vitamin B12 metabolism. They demonstrate that this mutation affects a splice acceptor site, the consequence of which is read-through transcription of PRDX1 into the MMACHC gene in an antisense orientation. Antisense transcription through the promoter of MMACHC is associated with transcriptional silencing of this promoter and hypermethylation of the promoter associated CpG island. There is also an increase in H3K36me3 around this site in individuals with this mutation. H3K36me3 is a mark that is associated with active transcription of PolII and is involved in splicing, supporting the other evidence of antisense read-through of the promoter of the neighbouring MMACHC (i.e. aberrant transcripts). They provide some functional evidence for this hypothesis by showing that knock down of the aberrant PRDX1 restores transcription of MMACHC. This mutation was discovered as when inherited as a compound heterozygote with loss of function mutations within the MMACHC coding region, it is sufficient to cause cblC metabolic disorder due to loss of MMACHC expression. They argue that this is an example of a constitutive epimutation that is transgenerationally inherited as the hypermethylation phenotype is present in sperm in cases with paternal transmission.*”

Novelty: These findings are novel in that this is a rare and novel mutation that is associated with an inherited disorder and has diagnostic value for cases of suspected cblC in which homozygous MMACHC mutations are absent. With regards to the claim of a transgenerationally inherited epimutation; I would apply caution. This is clearly a genetic mutation that links to phenotype through altering the epigenetic state, as they show. The presence of promoter hypermethylation in sperm does not demonstrate transgenerational epigenetic inheritance, (i.e. failure to be erased either in primordial germ cells and/or the post-fertilisation embryo).” The evidence they present suggests that hypermethylation will be established in any tissue in which PRDX1 is expressed (including the male germ-line during spermatogenesis) as an obligatory consequence of the genetic mutation. They provide no direct evidence to suggest that there is failure to erase this during the developmental periods of genome-wide DNA methylation erasure. Furthermore, this is not the first example of read-through antisense transcription altering the epigenetic state of a promoter. Therefore, with regards to mechanistic insight, this is not novel (e.g. Tufarelli, C. et al. Transcription of antisense RNA leading to gene silencing and methylation as a novel cause of human genetic disease. Nature Genet. 34, 157–165 (2003).)

I think the conclusions and work presented in the manuscript are reasonable. I have some minor comments with regards to specific analyses and presentation, provided below.

I believe the greatest relevance and interest in this work will be for the diagnostic benefit of this specific disorder, rather than an insight into novel biological mechanisms.”

We thank the reviewer for these general appreciations of our work on its limitations, novelty and relevance.

Concerning the following comment: *“With regards to the claim of a transgenerationally inherited epimutation, I would apply caution. This is clearly a genetic mutation that links to phenotype through altering the epigenetic state, as they show. The presence of promoter hypermethylation in sperm does not demonstrate transgenerational epigenetic inheritance, (i.e. failure to be erased either in primordial germ cells and/or the post-fertilisation embryo). “*, we would like to point out that we do not claim that the transmission was directly “inherited”. We have written that there is a transgenerational transmission of the epimutation, with the presence of the epimutation in three generations and its presence in sperm. In addition, we have written that the transmission is triggered by the antisense transcription of *PRDX1* produced by the *PRDX1* mutations, in one heading of the result section “The epimutation was generated by aberrant extension of anti-sense transcription of *PRDX1* through the *MMACHC* exon1 and the *MMACHC/CCDC163P* bidirectional promoter.” and in the second paragraph of the discussion: “The forced antisense transcription of *MMACHC* resulted from the skipping of the last exon of *PRDX1*. This skipping was the causative defect that produced the epimutation since the silencing of *PRDX1* decreased the methylation of exon 1 and the promoter and restored the transcription of *MMACHC*, in WG-3838 and MeWo-LC1 cells.”

In regard to the comment of the reviewer, it seems that we have to explain this point more clearly. To address this comment, we have revised the second sentence of the discussion as follows: “The epimutation is directly involved in the mechanism of the disease and is transmitted in 3 generations through the forced antisense transcription of the adjacent mutated *PRDX1*.”

We have also revised the abstract with the following sentence “The epimutation was transmitted in three generations through *PRDX1* mutations that forced antisense transcription of *MMACHC*, resulting in a H3K36me3 mark in the promoter.”

Concerning the following comment: *“The evidence they present suggests that hypermethylation will be established in any tissue in which PRDX1 is expressed (including the male germ-line during spermatogenesis) as an obligatory consequence of the genetic mutation.”* we would like to indicate that we agree with the reviewer. This is what we have tried to express in the discussion: “The transgenerational transmission and the presence of the *MMACHC* epimutation in sperm may be explained by the ubiquitous high expression of *PRDX1* in germ cells, stem cells, and somatic cells.” We have revised the sentence as follows, to make this point clearer: “The transgenerational transmission and the presence of the *MMACHC* epimutation in DNA from sperm, fibroblasts and blood may be explained by the ubiquitous high expression of *PRDX1* in germ cells, stem cells, and somatic cells.”

To address the following comment: *“They provide no direct evidence to suggest that there is failure to erase this during the developmental periods of genome-wide DNA methylation erasure.”*, we have added the following sentence in the third paragraph of discussion “We

demonstrated that the epimutation escaped spermatozoa erasure, in contrast to previous reports of other diseases^{5,14}, but we could not assess the possibility of a hypothetical failure to erase this epimutation during early embryonic development.”

Regarding the following comments: “*Furthermore, this is not the first example of read-through antisense transcription altering the epigenetic state of a promoter. Therefore, with regards to mechanistic insight, this is not novel (e.g. Tufarelli, C. et al. Transcription of antisense RNA leading to gene silencing and methylation as a novel cause of human genetic disease. Nature Genet. 34, 157–165 (2003).)*”, we would like to point out that we agree with the reviewer. The article by Tufarelli et al. in Nat. Genet. 2003 is one of the previous examples of gene silencing by methylation through a gene disruption that caused antisense transcription across the CpG island of the promoter. However, these authors reported the absence of methylation in sperm, and studied only two generations.

We have added a sentence of the third paragraph of discussion section to address this comment:

“Gene silencing by methylation through a gene disruption that caused antisense transcription across the CpG island of the promoter was previously reported for *HBA2* (alpha-thalassemia) and *MLH1* (familial cancer syndrome).”

More precisely, the initial genetic data of the cases of alpha-thalassemia were reported by Barbour, Tufarelli et al in a former article published in Blood, 2000. The Nat Genet paper cited by the referee corresponds to subsequent molecular studies in ES cells and transgenic mouse models. The report of these cases in the article of Blood indicates that the promoter was not methylated in sperm, as pointed out in our discussion. This lack of methylation is clearly indicated in page 803 of the article published in Blood: “the CpG island (H) associated with the remaining $\alpha 2$ gene on the $\alpha 2$ -ZF chromosome was unmethylated in spermatocytes (Figure 4C), but it was methylated in peripheral blood, EBV lymphocytes, and the interspecific hybrid”. In addition, the methylation of $\alpha 2$ gene was studied only in two generations, e.g. the proband, his affected mother and his unaffected father (table 1).

Specific remarks

Point 1: “*I found the manuscript difficult to read and perhaps some effort could be used to simplify language where possible, e.g. lines 279-281 “rare disease produced by compound epigenetic/genetic heterozygosity in a reverse1 (R1)/forward 2(F2)/reverse3(R3) trio of genes),”*”

We have tried to tighten up the language throughout the manuscript and we have simplified headings of paragraphs in the Results section. We have revised the sentence: “rare disease produced by compound epigenetic/genetic heterozygosity in a reverse/forward/reverse trio of genes)”

Point 2: “*-with the WGS, did you consider trans variants as well, if not, what do you consider a cis variant?”*”

As explained result section, we used an informative variant, rs3748643 to identify the position in cis of the *PRDX1* mutation and the epimutation. This is explained in the Results section for CHU-12122: “We detected a c.-302G genotype of the rs3748643 c.-302 G>T polymorphism in the allele bearing the epimutation and a c.-302T genotype in the non-methylated allele. The epimutation and c.-302G genotype of rs3748643 were absent in DNA from the mother and maternal grandmother (Fig. 1 a,c).” and also in the same section for WG-4152: “The c.-302G genotype of rs3748643 was detected in the allele bearing the epimutation and the c.-302T genotype in the non-methylated allele (Fig. 2b,c).” These positions according to rs3748643 were confirmed in WGS. We have reworded the corresponding sentence of the Results section to address the comment of the

reviewer: “The *PRDX1* variants and the polymorphism rs3748643 associated with the epimutation were present in the same allele as evidenced by DNA sequencing and transmission in the heterozygous relatives.”

Point 3: “*the layout of the figures is confusing. E.g. Figure 1 relates to one pedigree. Figure 2a also relates to this pedigree as does the top part of 2b, whereas the middle of 2b is the next pedigree and the bottom the other patient, before 2c goes on to relate to the second pedigree. Having figures correspond to the order they are mentioned in the text AND grouped into related information would make it easier for the reader.*”

We have revised the layout of the figures as suggested. Figure 1 relates only to data from CHU-1222 and her relatives. Figure 2a is now Figure 1d as it relates to CHU-12122. The middle of 2b and 2c are now Fig 2a and Fig 2b, respectively. They relate to WG-3838 and relatives. Fig 2 c, d, and e present the data of HM250 methylome profiling in the cases and their relatives.

Point 4: “*-Figure 6b; they use the wrong control*”

We used two types of controls in Fig 6b, control fibroblasts and absence of RNA extract in the RT-PCR. In lane 6, RT-PCR shows no detectable antisense RNA in RNA extracted from HDF control fibroblasts. When no RNA is added in the RT-PCR reaction mixture, no artifactual amplification is observed. We have revised the legend as follows: “Lanes 2, 4 and 6 correspond to RT-PCR of RNA from fibroblasts CHU-12122, WG-3838 and HDF (control fibroblasts). Lanes 3 and 5 correspond to control experiments without fibroblast RNA in the reaction mixture of RT-PCR. They show no artifactual amplification.”

Reviewer #3:

General comments: “*Methylmalonic aciduria and homocystinuria, cblC type, is rare recessive disease caused by biallelic loss of function of the MMACHC gene on chromosome 1. The authors initially identified three patients in which one MMACHC allele has a DNA sequence mutation and the other one an epimutation (promoter hypermethylation). In two families the epimutation is also present in the patient's fathers, and in at least one of them, it also appears to be present in sperm. Interestingly, the epimutation is in phase with a DNA sequence mutation in a neighbouring gene (PRDX1), which is transcribed towards the MMACHC gene from the other strand (convergent promoters). The authors show that the mutation leads to skipping of the last PRDX1 exon and the transcription termination signal, so that the polymerase reads – in antisense - through the MMACH1 gene into the neighbouring gene on the other side (CCDC163P). It is likely that PRDX1 read through transcription induces methylation of the MMACHC promoter. This notion is supported by the presence of H3K36me3 at the MMACHC promoter and by hypomethylation and reactivation of MMACHC expression after silencing of PRDX1. In a consecutive study, the authors identified five more cases with an PRDX1 mutation and MMACHC epimutation.*

In summary, Gueant et al. have identified an epimutation that is caused by a cis-acting DNA mutation. Similar cases involving read through transcription induced DNA methylation have been described before (e.g. HBA2 and MSH2), although in these cases aberrant methylation was not present in sperm. The authors claim that their study presents the first evidence for transgenerational transmission of an epimutation in humans.

For transparency I would like to state that I reviewed the manuscript previously for another journal. While the findings are very interesting and the paper has improved since the first submission, I have several points of criticism.”

We thank the reviewer for these general appreciations of our work and we address the “several points of criticism” in the following answers:

Point 1: *“Aberrant HBA2 and MSH2 methylation in the afore-mentioned cases is restricted to somatic tissues, because the neighbouring genes LUC7L and EPCAM, respectively, are expressed in a tissue-specific manner. In contrast, PRDX1 is expressed in all tissues including germ cells and stem cells. Germ cell expression of PRDX1 most probably explains why MMACHC methylation also occurs in sperm. Thus, we are dealing with the transgenerational transmission of a DNA (PRDX1) mutation that causes MMACH1 methylation in each cell, and not with the transgenerational transmission of an epimutation. The MMACHC epimutation per se is not stable, because it easily lost after blocking PRDX1 read through (Fig. 6; this also needs to be discussed more thoroughly). The term "transgenerational inheritance of an epimutation" is reserved for the inheritance of a (stable) epimutation that has occurred spontaneously in the absence of any DNA sequence change during the life of a parent. It is important to distinguish between these two different modes of inheritance, especially with respect to the discussion of the inheritance of acquired traits. Thus, the authors should not use the term "transgenerational inheritance of an epimutation" for their cases.”*

We agree with the reviewer that PRDX1 is expressed in all tissues including germ cells and stem cells and that germ cell expression of PRDX1 most probably explains why MMACHC methylation also occurs in sperm. This is indicated in the third paragraph of discussion, “The transgenerational transmission and the presence of the *MMACHC* epimutation in sperm may be explained by the ubiquitous high expression of *PRDX1* in germ cells, stem cells, and somatic cells.” and “*PRDX1* was also ubiquitously expressed in E7-E10 mouse embryos (<http://www.informatics.jax.org>; <http://dbtmee.hgc.jp/>) and adult humans (<http://www.proteinatlas.org>).” In contrast to the aberrant *HBA2* and *MSH2* methylation, we observed the aberrant methylation of *MMACHC* in somatic tissues and in sperm. We have revised the sentence as follows, to make this point clearer: “The presence of the *MMACHC* epimutation in DNA from sperm, fibroblasts and blood may be explained by the ubiquitous high expression of *PRDX1* in germ cells, stem cells, and somatic cells.”

Concerning the following comments: *“Thus, we are dealing with the transgenerational transmission of a DNA (PRDX1) mutation that causes MMACH1 methylation in each cell, and not with the transgenerational transmission of an epimutation. The epimutation is directly involved in the mechanism of the disease and is transmitted in 3 generations through the forced antisense transcription of the adjacent PRDX1 mutated gene.”* and *“The term "transgenerational inheritance of an epimutation" is reserved for the inheritance of a (stable) epimutation that has occurred spontaneously in the absence of any DNA sequence change during the life of a parent. It is important to distinguish between these two different modes of inheritance, especially with respect to the discussion of the inheritance of acquired traits. Thus, the authors should not use the term "transgenerational inheritance of an epimutation" for their cases.”*, we would like to point out that we do not claim that this transmission is directly inherited. This is the reason why we use the term “transgenerational transmission” instead of “transgenerational inheritance”. We have clearly written that the “transgenerational transmission” of the epimutation is triggered by the antisense transcription of *PRDX1* produced by the *PRDX1* mutations, in the second paragraph of the discussion: “The forced antisense transcription of *MMACHC* resulted from the skipping of the last exon of *PRDX1*. This skipping was the causative defect that produced the epimutation since the silencing of *PRDX1* decreased the methylation of exon 1 and the promoter and restored the transcription of *MMACHC*, in WG-3838 and MeWo-LC1 cells.”

To make this point clearer and to address the comments of the reviewer, we have revised the second sentence of the discussion as follows: “The epimutation is directly involved in the mechanism of the disease and is transmitted in 3 generations through the forced antisense transcription of the adjacent mutated *PRDX1*.”

We have also revised the abstract with the following sentence “The epimutation was transmitted in three generations through *PRDX1* mutations that forced antisense transcription of *MMACHC*, resulting in a H3K36me3 mark in the promoter.”

Regarding the comment on the *MMACHC* epimutation stability and its loss after blocking *PRDX1*, we would like to point out that our experimental data do not support that this epimutation is easily lost and unstable, considering that the silencing of *PRDX1* produced only 10-15% hypomethylation of the allele initially fully methylated. We have discussed this point in the second paragraph of discussion:

“However, the silencing of *PRDX1* produced only 10-15% hypomethylation of the allele initially fully methylated, suggesting a limited reversibility of the epimutation (Fig. 6g).

Point 2: “ *The authors claim that the epimutation is also present in sperm from the fathers of CHU-12122 (CHU-14061) and WG-3838 (CDH-867). However, MMACHC methylation and SNRPN methylation (contamination control) is only shown for CHU-14061. The MMACHC and SNRPN data for CDH-867 need to be shown, also.*”

We have now added these data in the revised figure 2b (Sanger of bisulfated DNA), Fig 2e (HM 450 methylome profiling) and in the supplementary figure S1 (methylation of SNRP imprinted gene in sperm from proband’s fathers and two controls).

Point 3. “*I am not convinced that the patients need to be reclassified as having a new type of Cbl inherited disorder, epi-cblC. I assume that MMACHC methylation mimics an MMACHC mutation and that the clinical features are the same.*”

We consider “epi-*cblC*” as a new cause for the *cblC* inherited disorder, but not as a new group of *Cbl* inherited disorders. This is now more clearly indicated to address the comment of the reviewer, in the summary and in the introduction section: “We report the transgenerational transmission of a new cause of the autosomal recessive *cblC* class of inborn errors of vitamin B12 metabolism that we named “epi-*cblC*”, the first heading of the Results section: ”Identification of a new cause of the *cblC* disorder named epi-*cblC*...” and the first sentence of the discussion : “Our “epi-*CblC*” cases represent a new cause for the autosomal recessive *cblC* disorder...”.

However, we cannot consider our epi-*cblC* cases as “classical” *cblC* patients as we do not know whether the *PRDX1* heterozygous mutations could have an influence on the clinical symptoms throughout life. In addition, it is important to distinguish the epi-*CblC* cases from the classical *cblC* patients for preimplantation genetic diagnosis. This was the case for the parents of case CHU-12122.

Point 4. “*Bisulfite sequencing of the MMACHC promoter in sperm DNA of 14061 should be shown in Fig 1, not Fig. 2.*”

We have modified the Figure 1 and Figure 2 as suggested, in addition to the changes indicated by reviewer 2.

Point 5: « *Terminology is sometimes wrong or imprecise. Examples for wrong terminology:*

Line 113 and ff: "*c.-302G genotype of the rs3748643 ...2*". This should read "*allele*" rather than "*genotype*". Lines 212-213 "*The level of anti-sense transcripts was high in control and case fibroblasts*". In the controls, there is no *MMACHC* antisense-transcription, just the sense *PRDX1* transcription. Lines 217-219: "*The activation of cryptic acceptor sites and resulting skipping of exons and introns of MMACHC ...*". The exons and introns are not skipped; they are not recognized, because they are on the other strand. Line 235: "*compound heterozygosity for a constitutional epimutation ... in cis and MMACC coding mutation in trans*." This is an inappropriate use of the words "*cis*" and "*trans*".

We have changed wording from "genotype" to "allele"

We thank the reviewer for pointing out the error in Lines 212-213. As shown in Fig 6b, the *MMACHC* antisense-transcription was undetectable in control fibroblasts. We have corrected the sentence as follows: "The level of anti-sense transcripts was high in case fibroblasts and undetectable in control fibroblasts."

Regarding the comment on Lines 217-219, the RNA seq experiments showed that the forced antisense transcription produced the activation of cryptic acceptor sites that generated several antisense *MMACHC* transcripts. We also identified some of these transcripts by RT-PCR.

We deleted "exons and introns" and we reworded the related sentences from the revised first paragraph of page 7, to address the comment:

"The forced antisense transcription produced the activation of cryptic acceptor sites that generated several antisense *MMACHC* transcripts" and "This transcript resulted from the activation of a cryptic antisense splicing site located in the middle of *MMACHC* exon 1 (Fig. 6d,e)."

To answer the comment on Line 235, we have deleted *trans* and *cis* and indicated that "Our *epi-Cbl* cases represent a new type of *cbIC* inherited disorder, resulting from hypermethylation of the CpG island in the bidirectional promoter of *MMACHC* on one allele and a coding mutation on the other."

We thank the reviewers for their help in improving the quality of our manuscript and we hope that it is now acceptable for publication.

Yours sincerely,

Pr Jean-Louis Guéant, MD, PhD, AGAF
Director of Inserm UMRS 954, (Nutrition-Genetics-Environmental Risks)
Institute of Medical Research (Pôle BMS) - University of Lorraine
Corresponding author

Professor David S. Rosenblatt, MD,
Department of Human Genetics, McGill University

Reviewer #2 (Remarks to the Author):

I am still concerned that the language used to describe their findings implies 'epigenetic inheritance.' Germ-line epigenetic inheritance in mammals is still highly controversial.

This is a case of inheritance of a genetic mutation that determines a functional consequence via epigenetic mechanisms to disrupt gene function. These mechanisms occur in both germ-line and somatic tissue.

I believe the work and the scientific community would both benefit from improved clarity.

Reviewer #3 (Remarks to the Author):

The authors have improved the manuscript, but one major issue and two minor issues remain.

1. The most important issue is the question of transgenerational epigenetic inheritance. As argued by Reviewer 2 and myself (Reviewer 3), the authors' claim is not justified. What the authors describe is a secondary epimutation of the MMACHC gene, i.e. an epimutation that is caused by an inherited mutation of the adjacent PRDX1 gene. In response to our critique, the authors have substituted "transgenerational INHERITANCE of an epimutation" by "transgenerational TRANSMISSION of an epimutation", which is not really a conceptual difference. In contrast to the previously described secondary epimutations of the MSH2 and HBA2 genes, for example, the MMACHC epimutation is present in sperm, too. This is probably due to the fact that the mutant PRDX1 transcript is not only expressed in somatic cells, but in germ cells also. However, the presence of the MMACHC epimutation in sperm is irrelevant for the inheritance and pathogenesis of the disease; it is the expression of the mutant PRDX1 transcript in the developing embryo that leads to silencing of one MMACHC allele and - if the other MMACHC allele is affected by a genetic mutation - to disease. The

terms "transgenerational inheritance/transmission of an epimutation" are reserved and must be reserved for the germ line transmission of primary epimutations that occur in the absence of a genetic mutation. In these cases the presence of the epimutation in sperm or oocytes is an absolute requirement for its transmission and phenotypic manifestation. The distinction between these situations is also important for the discussion on the transgenerational inheritance of acquired traits. The authors have tried to ameliorate their statement by adding "through PRDX1 mutations that forced antisense transcription of MMACHC" at several places in the manuscript, but this addition, although correct per se, blurs rather than clarifies the conceptual differences. In the abstract (line 35), for example, this leads to a sentence that doesn't make much sense: an epimutation cannot be "transmitted ... through ... mutations". It is the PRDX1 mutation that is transmitted through the germline, and this mutation causes the MMACHC epimutation in each cell in each generation. The authors need to correct their statement throughout the paper including the title. A possible title could be: "A MMACHC epimutation in germ cells and somatic cells is caused by readthrough transcription from a mutant PRDX1 allele". Admittedly, this is not as "sexy" as the original title, but closer to the truth.

Minor points:

2. The authors do not use the terms "cis" and "trans" correctly. "trans" refers to different loci, not to different alleles. The patients described in the manuscript are compound heterozygotes for a genetic mutation and an epigenetic mutation at the MMACHC locus. The PRDX1 mutation acts in cis on the MMACHC gene. This should be corrected throughout the manuscript.

3. The authors do not use the term "genotype" correctly. c.-302G and c.-302T, for example, are not genotypes, but alleles. This should be corrected throughout the manuscript.

Bernhard Horsthemke

We have tried to closely address the comments of reviewers #2 and #3. We have prepared point-by-point answers to the comments, and a revised version of the manuscript with the highlighted changes.

Reviewer #2:

"I am still concerned that the language used to describe their findings implies 'epigenetic inheritance.' Germ-line epigenetic inheritance in mammals is still highly controversial. This is a case of inheritance of a genetic mutation that determines a functional consequence via epigenetic mechanisms to disrupt gene function. These mechanisms occur in both germ-line and somatic tissue. I believe the work and the scientific community would both benefit from improved clarity."

Answer:

- We have deleted 'epigenetic inheritance' and "transgenerational transmission of the epimutation" throughout the manuscript

- We have rephrased the comment of reviewer #2 "This is a case of inheritance of a genetic mutation that determines a functional consequence via epigenetic mechanisms to disrupt gene function" in the second paragraph of the introduction section: "Here, we report cases with a rare inborn error of metabolism produced by inheritance of a gene mutation that disrupts the gene function of a flanking gene through epigenetic mechanisms in somatic and germ line cells"

- Horsthemke separates epimutations into two types, primary and secondary (reference 2 of our manuscript). We have introduced the definition of the secondary epimutation in the first paragraph of the introduction "Epimutations can be separated into two types, primary and secondary, the latter occurring secondary to a DNA mutation in a cis- or trans-acting factor^{1,2}" and we have used the terms "secondary epimutation" throughout the manuscript. We have changed the heading of the third paragraph of result section, which is now "The *MMACHC* epimutation occurs secondary to a DNA mutation in *PRDX1*". We have indicated in the second sentence of the discussion that "The secondary epimutation is caused by an inherited mutation of the adjacent *PRDX1* gene."

Reviewer #3:

"The authors have improved the manuscript, but one major issue and two minor issues remain.
1. *The most important issue is the question of transgenerational epigenetic inheritance. As argued by Reviewer 2 and myself (Reviewer 3), the authors' claim is not justified. What the authors describe is a secondary epimutation of the MMACHC gene, i.e. an epimutation that is caused by an inherited mutation of the adjacent PRDX1 gene. In response to our critique, the authors have substituted "transgenerational INHERITANCE of an epimutation" by "transgenerational TRANSMISSION of an epimutation", which is not really a conceptual difference. In contrast to the previously described secondary epimutations of the MSH2 and HBA2 genes, for example, the MMACHC epimutation is present in sperm, too. This is probably due to the fact that the mutant PRDX1 transcript is not only expressed in somatic cells, but in germ cells also. However, the presence of the MMACHC epimutation in sperm is irrelevant for the inheritance and pathogenesis of the disease; it is the expression of the mutant PRDX1 transcript in the developing embryo that leads to silencing of one MMACHC allele and - if the other MMACHC allele is affected by a genetic mutation - to disease. The terms "transgenerational inheritance/transmission of an epimutation" are reserved and must be reserved for the germ line transmission of primary epimutations that occur in the absence of a genetic mutation. In these cases the presence of the epimutation in sperm or oocytes is an absolute requirement for its transmission and phenotypic manifestation. The distinction between these situations is also*

important for the discussion on the transgenerational inheritance of acquired traits. The authors have tried to ameliorate their statement by adding "through PRDX1 mutations that forced antisense transcription of MMACHC" at several places in the manuscript, but this addition, although correct per se, blurs rather than clarifies the conceptual differences. In the abstract (line 35), for example, this leads to a sentence that doesn't make much sense: an epimutation cannot be "transmitted ... through ... mutations". It is the PRDX1 mutation that is transmitted through the germline, and this mutation causes the MMACHC epimutation in each cell in each generation. The authors need to correct their statement throughout the paper including the title. A possible title could be: "A MMACHC epimutation in germ cells and somatic cells is caused by readthrough transcription from a mutant PRDX1 allele". Admittedly, this is not as "sexy" as the original title, but closer to the truth."

Minor points:

2. The authors do not use the terms "cis" and "trans" correctly. "trans" refers to different loci, not to different alleles. The patients described in the manuscript are compound heterozygotes for a genetic mutation and an epigenetic mutation at the MMACHC locus. The PRDX1 mutation acts in cis on the MMACHC gene. This should be corrected throughout the manuscript.

3. The authors do not use the term "genotype" correctly. c.-302G and c.-302T, for example, are not genotypes, but alleles. This should be corrected throughout the manuscript.

Answer:

Point 1, regarding the following part of the comment: *"What the authors describe is a secondary epimutation of the MMACHC gene, i.e. an epimutation that is caused by an inherited mutation of the adjacent PRDX1 gene."*

We totally agree and we share this interpretation, which is a core message that we intended. The fact that the epimutation is secondary to a genetic mutation in the adjacent gene is central to our article. We investigated the existence of the gene mutation responsible of the epimutation by Whole Genome Sequencing and its consequences by RNAseq, Chip seq and cellular studies of invalidation of its expression. These results occupy a large part of the results section and more than half of the discussion. In the discussion we emphasize the association of the histone mark with de novo methylation. We illustrated our data by an explanatory scheme, which clearly shows that the epimutation is secondary to a mutation that forces antisense transcription of *PRDX1*. In a previous article, Horsthemke (Referee #3) has proposed to separate epimutations into two types, primary and secondary (reference 2 of our manuscript). According to this definition, the primary epimutations are those that occur in the absence of any DNA sequence change while the secondary epimutations are those that occur secondary to a DNA mutation in a cis- or trans-acting factor. To answer to this request of clarification, we have introduced the definition of the « secondary epimutation » in the first paragraph of introduction "Epimutations can be separated into two types, primary and secondary, the latter occurring secondary to a DNA mutation in a cis- or trans-acting factor" and we have used the term "secondary epimutation" throughout the manuscript.

Regarding the following part of the comment: *"In response to our critique, the authors have substituted "transgenerational INHERITANCE of an epimutation" by "transgenerational TRANSMISSION of an epimutation", which is not really a conceptual difference."*

Like reviewer #2, reviewer #3 is still concerned by any use of language that would imply 'epigenetic inheritance' to describe our findings. To address this comment, we have deleted the expression « transgenerational transmission of the epimutation » in the entire manuscript, and we replaced it by « presence of the epimutation in three generations », which is consistent with the data shown.

Regarding the following part of the comment: *"In contrast to the previously described secondary epimutations of the MSH2 and HBA2 genes, for example, the MMACHC epimutation is present in sperm, too. This is probably due to the fact that the mutant PRDX1 transcript is not only expressed in somatic cells, but in germ cells also."*

We agree with Reviewer #3. We have already covered this point in the manuscript in the third paragraph of the discussion “The presence of the *MMACHC* secondary epimutation in DNA from sperm, fibroblasts and blood may be explained by the ubiquitous high expression of *PRDX1* in germ cells, stem cells, and somatic cells.” and “*PRDX1* transcripts were detected throughout development, including in oocytes before and after maturation and in 2-c, 5-8c, 9-16c embryos, morulae, and blastocysts. *PRDX1* was also ubiquitously expressed in E7-E10 mouse embryos (<http://www.informatics.jax.org>; <http://dbtmee.hgc.jp/>) and adult humans (<http://www.proteinatlas.org>). “ To make this point more clear we have revised sentences related to the presence of the epimutation in sperm in the discussion section, first paragraph: “The high expression of *PRDX1* in spermatogenic cells may explain the presence of the epimutation in sperm, in contrast to the spermatozoa erasure previously observed in families with epigenetic silencing of the *MLH1* gene^{5,14}.” and third paragraph: “The high expression of *PRDX1* in germ cells may explain why the secondary epimutation escaped spermatozoa erasure, in contrast to previous reports of other diseases^{5,14}. The high expression of *PRDX1* could also maintain this epimutation during early embryonic development.”

Regarding the following part of the comment “*The authors have tried to ameliorate their statement by adding “through PRDX1 mutations that forced antisense transcription of MMACHC” at several places in the manuscript, but this addition, although correct per se, blurs rather than clarifies the conceptual differences. In the abstract (line 35), for example, this leads to a sentence that doesn’t make much sense: an epimutation cannot be “transmitted ... through ... mutations”.*”

We have rephrased “through *PRDX1* mutations” by “The secondary epimutation is caused by an inherited mutation of the adjacent *PRDX1* gene” in the first paragraph of discussion. The sentence of the abstract has been also corrected “The epimutation is present in three generations and results from *PRDX1* mutations that force antisense transcription of *MMACHC* and produce a H3K36me3 mark. ”

Regarding the following part of the comment “*The authors need to correct their statement throughout the paper including the title. A possible title could be: “A MMACHC epimutation in germ cells and somatic cells is caused by readthrough transcription from a mutant PRDX1 allele”. Admittedly, this is not as “sexy” as the original title, but closer to the truth.*”

We have changed the former title by the following title, which takes into account the remark and indicates the type of disease in question, respecting the limit of 15 words of author’s instructions: « A *PRDX1* mutant allele causes an *MMACHC* secondary epimutation in *cb1C* patients »

Minor points:

Point 2. “*The authors do not use the terms “cis” and “trans” correctly. “trans” refers to different loci, not to different alleles. The patients described in the manuscript are compound heterozygotes for a genetic mutation and an epigenetic mutation at the MMACHC locus. The PRDX1 mutation acts in cis on the MMACHC gene. This should be corrected throughout the manuscript.*”

We have corrected this use in the abstract: “Subjects are compound heterozygotes for a genetic mutation and a promoter epimutation detected in blood, fibroblasts and sperm at the *MMACHC* locus” and throughout the manuscript, including in introduction “The epi-*cb1C* cases are compound heterozygotes for a genetic mutation and an epimutation at the *MMACHC* locus”, results “case CHU-12122, who was compound heterozygote for a genetic mutation and an epigenetic mutation at the *MMACHC* locus.” and conclusion “In conclusion, we report the first cases of a rare metabolic disease produced by compound heterozygosity of a secondary epimutation detected in somatic cells and sperm in one allele and a genetic mutation in the other.”

Point 3. *“The authors do not use the term “genotype” correctly. c.-302G and c.-302T, for example, are not genotypes, but alleles. This should be corrected throughout the manuscript.”*

This also has been corrected throughout the manuscript

We thank the reviewers for their comments on our revised manuscript and we hope that it will be now acceptable for publication.

Reviewer #2 (Remarks to the Author):

I am sufficiently satisfied that the clarity of the manuscript has been improved to prevent confusion around the molecular basis of the phenomenon described.

Reviewer #3 (Remarks to the Author):

The authors have resolved all of the remaining issues. Most importantly, they no longer use the term "transgenerational epigenetic inheritance". Furthermore, they have corrected the inappropriate use of the terms "genotype" and "cis/trans". As I mentioned before, this is a very interesting piece of work.

B. Horsthemke

Point-by-point answer to reviewer's comments:

Reviewer #2 (Remarks to the Author):

I am sufficiently satisfied that the clarity of the manuscript has been improved to prevent confusion around the molecular basis of the phenomenon described.

Answer: We thank the reviewer for her/his positive comment

Reviewer #3 (Remarks to the Author):

The authors have resolved all of the remaining issues. Most importantly, they no longer use the term "transgenerational epigenetic inheritance". Furthermore, they have corrected the inappropriate use of the terms "genotype" and "cis/trans". As I mentioned before, this is a very interesting piece of work.

B. Horsthemke

Answer: We thank the reviewer for his appreciation of the revised manuscript